# Mapping schistosomiasis risk landscapes and implications for disease control: A case study for low endemic areas in the Middle Paranapanema river basin, São Paulo, Brazil

Vivian Alessandra Ferreira da Silva[1☯]*, Milton Kampel[1,2,3☯], Rafael Silva dos Anjos[4‡], Raquel Gardini Sanches Palasio[5‡], Maria Isabel Sobral Escada[1,4‡], Roseli Tuan[6‡], Alyson Singleton[7‡], Caroline Kate Glidden[8‡], Andrew Chamberlin[9‡], Giulio Alessandro De Leo[9,10‡], Adriano Pinter dos Santos[6‡], Antônio Miguel Vieira Monteiro[1,2,4☯]*

**1** Remote Sensing Postgraduate Program (PGSER), Coordination of Teaching, Research and Extension (COEPE), National Institute for Space Research (INPE), São José dos Campos, São Paulo, Brazil, **2** Earth Observation and Geoinformatics Division (DIOTG), General Coordination of Earth Science (CG-CT), National Institute for Space Research (INPE), São José dos Campos, Brazil, **3** MOceanS - Monitoring Oceans from Space Laboratory, National Institute for Space Research (INPE), São José dos Campos, SP, Brazil, **4** LiSS – Laboratory for investigation of Socio Environmental Systems, National Institute for Space Research (INPE), São José dos Campos, SP, Brazil, **5** Laboratory of Spatial Analysis in Health (LAES), Department of Epidemiology, School of Public Health, University of Sao Paulo (FSP/USP), São Paulo, SP, Brazil, **6** Pasteur Institute, São Paulo, SP, Brazil, **7** Emmett Interdisciplinary Program in Environment and Resources, Stanford University, Stanford, California, United States of America, **8** Department of Biology, Stanford University, Stanford, California, United States of America, **9** Department of Oceans, Hopkins Marine Station, Stanford University, Pacific Grove, California, United States of America, **10** Woods Institute for the Environment, Stanford University, Stanford, California, United States of America

☯ These authors contributed equally to this work.
‡These authors contributed equally to this work.
* vivianafdsilva@gmail.com (VAFS); miguel.monteiro@inpe.br (AMVM)

## Abstract

### Background

Schistosomiasis, a chronic parasitic disease, remains a public health issue in tropical and subtropical regions, especially in low and moderate-income countries lacking assured access to safe water and proper sanitation. A national prevalence survey carried out by the Brazilian Ministry of Health from 2011 to 2015 found a decrease in human infection rates to 1%, with 19 out of 26 states still classified as endemic areas. There is a risk of schistosomiasis reemerging as a public health concern in low-endemic regions. This study proposes an integrated landscape-based approach to aid surveillance and control strategies for schistosomiasis in low-endemic areas.

### Methodology/Principal findings

In the Middle Paranapanema river basin, specific landscapes linked to schistosomiasis were identified using a comprehensive methodology. This approach merged remote sensing, environmental, socioeconomic, epidemiological, and malacological data. A team of experts identified ten distinct landscape categories associated with varying levels of

**Data Availability Statement:** All relevant data are within the manuscript and its Supporting information files.

**Funding:** This study was financed in part by the Coordenação de Aperfeiçoamento de Pessoal de Nível Superior - Brasil (CAPES) - Finance Code 001. This study was also funded by the Regular Research Project, Integrated risk mapping and targeted snail control to support schistosomiasis elimination in Brazil and Cote d'Ivoire under future climate change, Awarded n. 2019/23593-3, FAPESP-SP/Brazil Line of Funding associated with the BELMONT FORUM Cooperation Agreements. Field work was partially funded by a seed grant of the Stanford Center for Innovation in Global Health (SPO#231988, GL). VAFS was funded by CAPES-Coordenação de Aperfeiçoamento de Pessoal de Nível Superior, Ministry of Education of Brazil (CAPES, Finance Code 001) and the AEB - Brazilian Space Agency, with a Master's Scholarship for the graduate programme in Remote Sensing and Geoinformation at INPE. MK has been partially funded by grants from UKRI-GCRF 723 (EP/T003820/1), FAPESP (2021/04128-8), and FUSP (2017/00686-0). AP received founding from FAPESP/Belmont Forum, (2019/23593-3). GADL, ALS, CG and AC have been partially supported by US NSF (grant ICER-2024383 and DEB – 2011179) and by a seed grant of the Stanford Center for Innovation in Global Health. AS, CG, AC and GL were partially funded by the Belmont Collaborative Forum for Climate, Environment and Health (2019/23593-3) and the US National Science Foundation (grant ICER-2024383 and DEB – 2011179). The funders had no role in study design, data collection and analysis, decision to publish, or preparation of the manuscript.

**Competing interests:** The authors have declared that no competing interests exist.

schistosomiasis transmission potential. These categories were used to train a supervised classification machine learning algorithm, resulting in a 92.5% overall accuracy and a 6.5% classification error. Evaluation revealed that 74.6% of collected snails from water collections in five key municipalities within the basin belonged to landscape types with higher potential for *S. mansoni* infection. Landscape connectivity metrics were also analysed.

## Conclusions/Significance

This study highlights the role of integrated landscape-based analyses in informing strategies for eliminating schistosomiasis. The methodology has produced new schistosomiasis risk maps covering the entire basin. The region's low endemicity can be partly explained by the limited connectivity among grouped landscape-units more prone to triggering schistosomiasis transmission. Nevertheless, changes in social, economic, and environmental landscapes, especially those linked to the rising pace of incomplete urbanization processes in the region, have the potential to increase risk of schistosomiasis transmission. This study will help target interventions to bring the region closer to schistosomiasis elimination.

### Author summary

Schistosomiasis is a Neglected Tropical Disease whose transmission in Brazil is related to human contact with water contaminated by the trematode parasite *Schistosoma mansoni*. The national prevalence survey from 2011–2015 revealed a decline in schistosomiasis positivity rates to 1%, marking areas in the Middle Paranapanema river basin in SP-Brazil (MP) as low endemic. However, control programs face additional challenges due to the new dynamics connected with the social, environmental, and economic developments fueled by an urban-industrial urbanization model in the MP region. To address this, our study proposes a landscape-based methodological approach that categorizes and assesses schistosomiasis transmission potential in different regional landscape patterns. By combining remote sensing, environmental, socioeconomic, epidemiological, and malacological data, a multidisciplinary team identified ten distinct landscape-unit types associated with varying levels of schistosomiasis transmission potential. Using a decision-tree based machine learning classification method, we mapped schistosomiasis risk across the basin at a landscape scale. The identification of landscape features associated with schistosomiasis transmission risk will help to fine tune suitable surveillance and control strategies at both local and regional levels. Our findings indicate that landscape-units with higher transmission potentials are less interconnected across the basin. Low connectivity at the landscape scale, in part, explains the MP low endemic areas, contributing to the low endemicity in the Middle Paranapanema region. This approach offers promise for local and regional schistosomiasis management efforts aimed at eradicating the disease.

## Introduction

Schistosomiasis is a Neglected Tropical Disease (NTD) and one of the most widespread parasitoses in the world, registered in Africa, Asia, the Caribbean and South America. The World Health Organization (WHO) Fact sheets for schistosomiais has estimated that, at least, 251.4 million people required preventive treatment in 2021 [1]. Human schistosomiasis is an acute

and chronic water-based disease that presents itself in two major forms, intestinal and urogenital, and it is prevalent in tropical and subtropical areas, especially in low and moderate-income countries lacking assured access to safe water and proper sanitation [24]. It is caused by parasitic trematodes from the genus *Schistosoma* which has 5 of its main species displaying distinct geographical distributions around the globe. The parasite matures within aquatic snails, acting as intermediate hosts until it transforms into a stage called cercariae, capable of infecting humans who come into contact with contaminated water sources. The etiologic agent in Brazil is the *Schistosoma mansoni* Sambon, 1907, where three species of snails of the genus *Biomphalaria* have been found to be naturally infected: **B. glabrata** (Say, 1818), **B. straminea** (Dunker, 1848) and **B. tenagophila** (d'Orbigny, 1835). These snails are widespread throughout the country [2].

The initial cases of schistosomiasis were documented in Brazil in 1908. In the years that followed until the early 1930s, investigations into the geographical distribution of the disease intensified and clinical, parasitological and malacological data were accumulated, showing that schistosomiasis was an important public health problem. In the 1950s, schistosomiasis was consolidated as an endemic disease in Brazil, mainly prevalent in regions characterized as rural areas [3, 4]. The Schistosomiasis Control Programme (PCE), initiated in Brazil in 1975, introduced uniform and integrated measures across the country, focusing on the reduction of disease prevalence and its morbidity in endemic areas by employing praziquantel treatment and molluscicide (Bayluscid) application in freshwater collections with *S. mansoni*-infected snails.

The prevalence of the disease has significantly declined from double-digit figures in the 1950s to a range of 1% to 5% in recent years. This indicates the effectiveness of the Schistosomiasis Control Program (PCE) strategy and the advancements in access to water, sanitation, and hygiene (WASH) in Brazil in altering the disease's prevalence [5, 6]. However, the risk of schistosomiasis transmission persists in specific geographical areas, now, different from the past, often situated near urban areas or peri-urban zones. These new urbanised areas and the zones between these areas of greater human population of built environment density and the rural areas, with lower population and concentrations, which we refer to here as peri-urban areas, are spaces characterised by an incomplete urbanisation where production spaces and housing are established, but with inadequate access to sewage treatment and services that would prevent the contamination of water by infected snail [7–11].

A substantial shift in the distribution patterns of *B. straminea*, *B. glabrata*, and *B. tenagophila* was documented in a malacological investigation conducted within freshwater collections situated in the middle section of the Paranapanema river basin in Sao Paulo, Brazil [12]. This shift may reflect fluctuations in the capacity of specific habitats to sustain these species, but the potential impact of these changes on disease transmission is still largely unknown and has received so far very little attention on the scientific literature.

Additionally, the three species display varying responses to changes in surface water temperature [13]. As the global temperature continues to rise, there is a likelihood of an increase in the frequency and intensity of extreme weather events. Changes in water temperature, shifts in rainfall patterns, and the occurrence of severe flooding leading to the expansion of flooded areas could be linked to the emergence of this new climate stage. Changing climate brings new challenges for the fight against schistosomiasis [14, 15]. Land use and land cover changes have transformed the middle section of the Paranapanema river basin at the landscape scale, impacting the basin's drainage system throughout the period of human occupation. This process has continued in recent years. These circumstances novel dynamics within the basin's waterways, that may connect freshwater habitats that currently host these species with areas

where they were previously absent. This scenario remains relatively little explored in current literature [16, 17].

Two recent WHO documents dealing with neglected tropical diseases in general and schistosomiasis in particular, put the elimination of schistosomiasis as a public health problem at the centre of the agenda [18–20]. Although there has been forward progress towards this goal, we are still reliant on mass administration of praziquantel (MDA) as our main control strategy. A new pediatric formulation of this chemotherapy drug for children between 3 months and 6 years old is expected to be soon available on the market [21]. Also, although there has been good progress in WASH strategies, education and communication actions about the disease, there is still an uneven distribution of these improvements across the different regions of the planet [5].

For areas of low endemicity, such as the Middle Paranapanema basin, these new scenarios of possible changes in rainfall and temperature patterns associated to changes in land use, urban expansion, and movements of people and goods, pose major challenges for schistosomiasis control and elimination programmes in the wake of advances in the processes of a metropolitan model of urbanisation in the state of São Paulo towards small and medium-sized cities. In these areas, it is essential to think of new strategies for mapping areas at risk of potential transmission that can guide targeted intervention actions. In these transformed and changing landscapes, the relationships and interdependencies that snails, humans and parasites have with each other are also altered to varying degrees. Therefore, thinking about control and elimination in situations of low endemicity requires a focus on a geographic region of action and the characterization of its specific landscapes for the contexts involving schistosomiasis and its transmission. However, considering the connectivity of these local landscapes, the relationship of these landscapes with other schistosomiasis landscapes on a regional scale is essential. To address schistosomiasis low endemic areas transmission potential risk mapping oriented towards surveillance and control systems involved in strategies for disease control and elimination programs, our study proposes a landscape-based method approach [22, 23].

Our study fits within a methodological framework that categorizes and assesses schistosomiasis transmission potential for different regional landscape patterns. These patterns were defined by a multidisciplinary team, which combined local knowledge and literature with orbital remote sensing images, environmental, socioeconomic, epidemiological and malacological data. Our approach offers promise for local and regional schistosomiasis management efforts that seek its elimination by 2030 by promoting surveillance-response systems instruments that are more tailored to the social- ecological settings found in the basin region.

## Methodology

### Study area

The Middle Paranapanema river basin (MP), located in Southeast, Brazil, includes 55 municipalities. The region stands out among the endemic sites for schistosomiasis in the state of São Paulo, Brazil, with a transmission history since 1952. The first autochthonous cases occurred in the municipalities of Ourinhos, Ipaussu and Palmital [25, 26]. Eight out of 55 municipalities in the MP region have reported autochthonous cases of schistosomiasis. The last autochthonous cases in the MP region occurred in the municipalities of Ourinhos (2018), Ipaussu and Ribeirão do Sul (2017), Chavantes and Santa Cruz do Rio Pardo (2016), Palmital (2011) and Canitar (2008) [27]. Although Assis had the second-highest number of cases within the historical record, trailing only behind Ourinhos, the occurrence of its last autochthonous cases dates back to 1997 [28]. These eight municipalities and another 17 municipalities in the MP basin are included as geosentinel units and are prioritised for schistosomiasis monitoring and

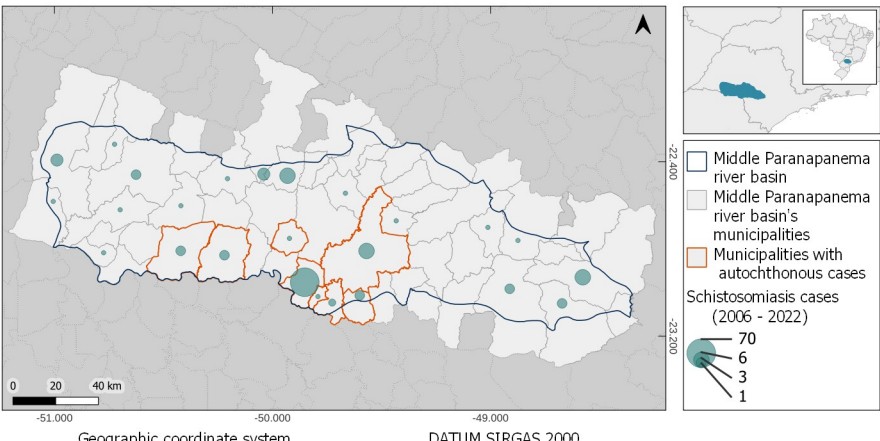

**Fig 1. The Middle Paranapanema river basin, São Paulo, Brazil, highlighting eight municipalities that have had a history of autochthonous cases and the aggregate of schistosomiasis cases by year of the onset of symptoms for the 55 municipalities in the basin in the period between 2006 and 2022.** Municipality border shape available from https://geoftp.ibge.gov.br/organizacao_do_territorio/malhas_territoriais/malhas_municipais/municipio_2022/Brasil/BR/BR_Municipios_2022.zip. Terms of use available from https://biblioteca.ibge.gov.br/visualizacao/livros/liv101998.pdf. MP border shape available from https://datageo.ambiente.sp.gov.br/geoserver/datageo/LimiteUGRHI/wfs?version=1.0.0&request=GetFeature&outputFormat=SHAPE-ZIP&typeName=LimiteUGRHI. License information available from https://datageo.ambiente.sp.gov.br/sobre.

control by the Epidemiological Surveillance Groups (GVE). The GVEs are part of the surveillance and control structure of the São Paulo state health department, and work with national programmes [29]. For the region of this study, the municipalities with autochthonous cases are part of the geosentinel units associated with GVE-13 (GVE Assis). The total population of the MP basin was 1,320,973 inhabitants in 2022 (IBGE), and the population of the eight municipalities with registered autochthonous cases was 302,316 inhabitants [32]. Epidemiological data on schistosomiasis cases in the MP basin region are available by the year of onset of symptoms (or by the year of notification of the disease) and aggregated at municipal level. The case database covers the period from 2006 to 2022 [30]. Fig 1 shows a map of the MP basin and highlights both the eight municipalities that have reported autochthonous cases and the aggregated number of schistosomiasis cases for the municipalities in the basin between 2006 and 2022. The numerical data used in this and the other figures are included in S1 Data.

## Data and methods

Fig 2 illustrates a concise schematic diagram detailing the fundamental procedural stages necessary for producing the schistosomiasis landscape risk maps. Within the context of the MP basin, the employed datasets are presented, in particular, regarding malacological and landscape characterization data. The methodology section delineates the process involved in constructing the typology of landscape units associated with schistosomiasis within the basin. Additionally, it elaborates on the methods employed to create the essential variables required for the classification process using machine learning techniques. Finally, it outlines how these variables were employed in a supervised classification strategy, utilizing decision tree methods.

## Malacological data

A comprehensive investigation into *Biomphalaria* identification was carried out between 2014 and 2018, sweeping a large set of water collection, across five municipalities that reported

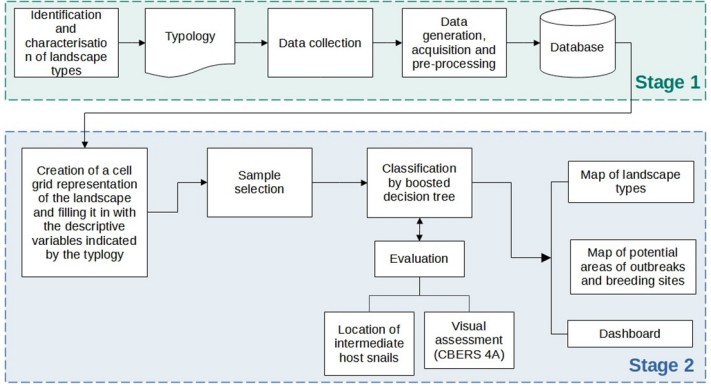

**Fig 2. Simplified general schematic diagram of methodology.**

autochthonous cases: Ourinhos, Ipaussu, Ribeiro do Sul, Chavantes, and Assis. The database relating to this collection, with the identification and localisation of three species of intermediary host snail, *B. glabrata*, *B. tenagophila* and *B. straminea*, has been used in this study [31]. Fig 3 shows their geographical distribution. According to the findings of Palasio et al. (2019) [31], every Biomphalaria sample gathered in the field tested negative for S. mansoni cercariae in laboratory.

## Landscape characterisation data

**Physical landscape data.** The land use and land cover (LULC data) utilized in this study was derived from the MapBiomas project [32]. The classes of agriculture, pasture, urbanised

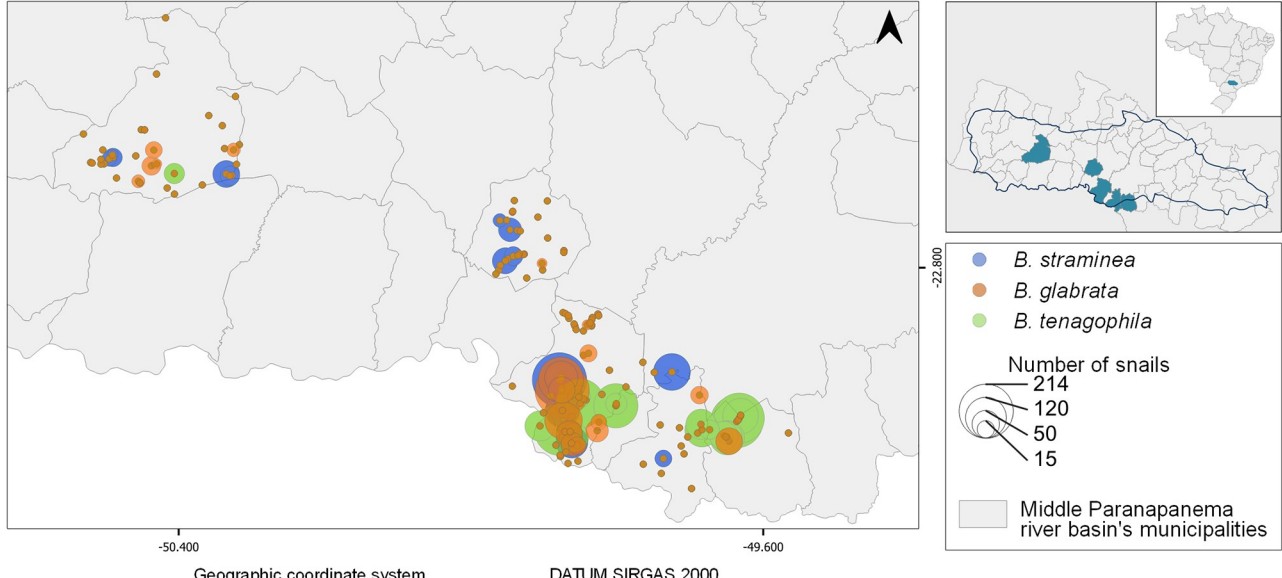

**Fig 3. *B. glabrata*, *B. tenagophila* and *B. straminea* geographical distribution in five municipalities of the MP basin: Ourinhos, Ipaussu, Ribeirão do Sul, Chavantes, and Assis.** Municipality border shape available from https://geoftp.ibge.gov.br/organizacao_do_territorio/malhas_territoriais/ malhas_municipais/municipio_2022/Brasil/BR/BR_Municipios_2022.zip. Terms of use available from https://biblioteca.ibge.gov.br/visualizacao/livros/ liv101998.pdf. MP border shape available from https://datageo.ambiente.sp.gov.br/geoserver/datageo/LimiteUGRHI/wfs?version=1.0.0&request= GetFeature&outputFormat=SHAPE-ZIP&typeName=LimiteUGRHI. License information available from https://datageo.ambiente.sp.gov.br/sobre.

areas and forest fragments (Atlantic forest) have helped to delineate landscapes with economic activity and human presence at different levels of potential contact with host snails, landscapes with a strong and more dense human presence and spaces with few chances of encounter between snails and humans respectively. Despite the inherent constraints of the 30-meter spatial resolution (Landsat images are used by MapBiomas), which may hinder the precise identification of targets demanding finer details, it proved suitable for the landscape scale applied in this particular investigation.

The roads database provided by Open Street Map (OSM) [33] was reclassified according to a Mobility Typology [34]. This reclassification facilitated the rapid identification of road types —whether local or regional—and their respective travel speeds. Distinguishing between local roads, residential pathways, non-automotive routes, and potential forthcoming infrastructural developments aided in pinpointing landscape types that exhibit higher potential for human-parasite interaction.

Phase Array L-band Synthetic Aperture Radar (PALSAR) imagery from the Advanced Land Observing Satellite (ALOS) [35] was used to derive several key elements. These included the relief-adjusted drainage network, High Above Nearest Drainage (HAND) calculations [36], and the assessment of the slope along the edges of water bodies. The generation of the relief-adjusted drainage network took into account the landforms delineated in the cartographic base of geomorphological units in Brazil [37]. This drainage network was created using an open source and free software, the TerraHidro platform [38], employing dual thresholds tailored to regions characterized by either high or low densities of water channels. Subsequently, the slopes along the margins of water bodies were computed based on the PALSAR/ALOS image mosaic associated with the relief-adjusted drainage network.

The High Above Nearest Drainage (HAND) method leverages the directional flow of nearby watercourses to compute the vertical distance from a specified location to the closest point on the drainage network. This set of nearest vertical distances was taken to identify regions contiguous to watercourses as potential habitats for intermediate host snails, drawing upon malacological data. Unlike "buffer" zoning operations which focus on horizontal distances between a point and the drainage, HAND accounts for terrain altitude fluctuations, offering a more nuanced assessment of topographical variations.

**Socio-demographic landscape data.** The Brazilian Demographic Census survey data provides information on households and their surroundings aggregated by spatial units called census tracts. Census tracks are classified by sector, i.e., as urban or rural, and for each of them the number of households is reported, as wall the situation of access to water and sanitation. We used the data from the 2010 census, the last one with disaggregated information at census tract level [39]. However, there is an updated sector mesh for the 2022 census [40] which only gives information on the sector's situation (urban or rural). This information, updated for 2022, better describes the urban/rural environment in the basin region today and was therefore used in this study.

The Rural Environmental Registry (CAR) stands as a mandatory national registry encompassing all rural properties in Brazil. Functioning as a self-declaratory system, its primary objective is to compile an integrated repository of information regarding rural properties and their alignment with the country's environmental data [41]. Within this study, CAR data served as an essential auxiliary information for refining the land use and land cover data class related to agricultural activities, based on the size of the declared rural property in the MP basin region. Notably, identifying smallholdings predominantly engaged in small-scale agriculture, relying on manual labour directly involved in land cultivation, helps to identify landscapes where the potential for contact between humans and the intermediate host snails of schistosomiasis. Conversely, medium and large rural properties, where production involves a

high degree of mechanisation, the use of chemical pesticides, and fewer rural workers, are landscapes with less potential for encounters between humans and snails. In order to use the CAR database, it was necessary to use a methodology to correct overlaps and to define property sizes [42].

Information on the presence of domestic effluents discharge, including black waters contaminated with fecal matters, was derived from the application of a methodology based on the probability of human waste concentration in the drainage network. For modelling the possibility of domestic effluents in the drainage system, we used the Digital Elevation Model of the ALOS sensor on board the PALSAR satellite, as well as data from the 2010 Demographic Census of the Brazilian Institute of Geography and Statistics (IBGE) and the location of sewage treatment plants provided by the National Water and Basic Sanitation Agency (ANA). The estimate of water pollution was based on the relationship between the number of inhabitants per census tract associated with the nearest drainage. Drainages without the presence of domestic effluents were considered to be rivers and streams that had census tracts without inhabitants or were downstream from sewage treatment plants and, during their course, did not receive any discharge from a river or stream with the potential presence of domestic effluents [43]. Identifying water bodies with untreated effluent discharges helps to indicate the areas most likely to be infected with *S. mansoni* by intermediate host snails.

The mosaic of uses class present in the land use and land cover database is a class that contains mixed agricultural uses related to small patches with vegetable production (e.g. lettuce, which is very common in the region), vegetable gardens and dirty pastures. Due to the 30 m spatial resolution used by the MapBiomas project, areas that cannot be properly classified into a specific type of agricultural production are included in this class, which states the presence of a mosaic of diversified small-scale agricultural uses.

In an approach based on the empirical exploration of data in the basin region, it was verified that 87% of autochthonous cases of schistosomiasis, reported between 1979 and 2016, were concentrated within a range of up to 225 meters from the contact areas between urbanised areas and the mosaic of uses class, in Ourinhos, the only municipality where finer spatial resolution cases data were available. Specifically, the threshold corresponds to the maximum distance between the boundaries of the urbanized areas and the aggregate cases of schistosomiasis per household, at census tract level, in the period 1979–2016 [27]. Using this threshold, a contact zone between urbanized areas and the mosaic of uses class were defined using buffers of 225 m from the polygons of the urbanized areas.

The data incorporated in this study (Table 1) were specifically chosen for their capacity to elucidate and delineate the various landscape typologies potentially linked with schistosomiasis within the research area.

## Typology building

Initially, in order to analyze schistosomiasis at the MP basin landscape scale, a multidisciplinary team delineated a set of schistosomiasis landscape patterns. These patterns were established based on the correlation observed between the constituents of the landscape being studied and their spatial arrangement in relation to the components involved in the transmission cycle of schistosomiasis. This team leveraged local expertise, field-gathered information, relevant literature, orbital remote sensing imagery, as well as environmental, socioeconomic, and malacological data to fully understand the landscape structures linked to schistosomiasis. These datasets were examined at fine spatial resolutions, some of which included geographical coordinates acquired via GPS. Epidemiological data on cases and incidence, available solely at the municipal level, were incorporated as an information to characterize this set of landscape

**Table 1. Type of information, derived variables/index, data sources, and the respective year of reference of the databases used in this study.** Data sources include IBGE (Brazilian Institute of Geography and Statistics), SICAR (National Rural Environmental Registry System), PALSAR/ALOS satellite data, and MapBiomas (annual land use and land cover maps across Brazil).

| Information | Derived variable/Index | Data source | Year of reference |
|---|---|---|---|
| Land use and land cover (LULC) classification | LULC classes; contatc between the urbanized area LULC class and the mosaic of uses LULC class | MapBiomas | 2022 |
| Situation of the census tract | Urban/Rural | IBGE | 2022 |
| Households | Household's presence | IBGE | 2010 |
| Agricultural properties | Presence of agricultural production in small rural properties / Presence of agricultural production in medium or large rural properties | SICAR | 2021 |
| Roads | Types of roads associated to a mobility profile | Open Street Map | 2021 |
| Edges slope | Slope of water body edges | PALSAR/ALOS images | 2011 |
| Drainage network | Drainage network; High above nearest drainage (HAND) and water body edges | PALSAR/ALOS images | 2011 |
| Water without proper treatment | Sections of the basin's drainage network with the presence of domestic effluent discharge. | IBGE, PALSAR/ALOS images | 2011 |

types. This set of types establishes a typology for the schistosomiasis landscape units patterns in the context of the MP basin. The typology seeks to represent the ecological characteristics of the intermediate snails hosts' habitat together with features related to socio-territorial aspects of the human population living in the MP basin region.

Based on the joint analysis carried out by the multidisciplinary team using field experience and the collected dataset made available in an integrated environment using geographic information system (GIS), ten distinct landscape-unit types associated with varying levels of schistosomiasis transmission potential in the MP basin were identified. The complexity of schistosomiasis transmission, which involves an etiological agent, intermediate host, and definitive host, was considered in the typology. The built typology indicates the variables that can describe the particularities of schistosomiasis landscape types and bases the landscape classification process on the scale of the MP basin.

Types 1 and 2 represent environments that are more prone for the establishment of the transmission cycle of schistosomiasis as they combine important elements that favour contact between the etiologic agent, the intermediate host, and the definitive host. Types 3 and 4 bring together important characteristics that support the fixation of intermediate host snails, especially the availability of freshwater. Although the areas of these types may support the development of human activities, they do not have a discharge of untreated domestic effluents into water bodies, which makes the landscapes less prone to contamination by *S. mansoni*. Types 5 to 10 are not prone to the settlement of intermediate host snails and, except for Type 10, are not characterized by intense and abundant human presence. In Types 5, 6 and 7, the proximity of drainage is insufficient to provide life support for the mollusks. In Types 8, 9 and 10, the absence of fresh surface water and the distance from water bodies also prevent the presence of *S. mansoni* and the snails *B. glabrata*, *B. straminea* and *B. tenagophila*.

The final typology built on the basis of these procedures can be seen in the supporting information material (S1 Fig) where each schistosomiasis landscape unit is characterised and the rationale for its definition is explained. In Fig 4 we can see an example for a full description of a landscape-unit of Type 1.

## Landscape gridded representation

The entire extent of the MP basin was represented as a grid of 500x500-meter cells, forming a standardized grid where each cell denotes a distinct landscape unit to be identified and

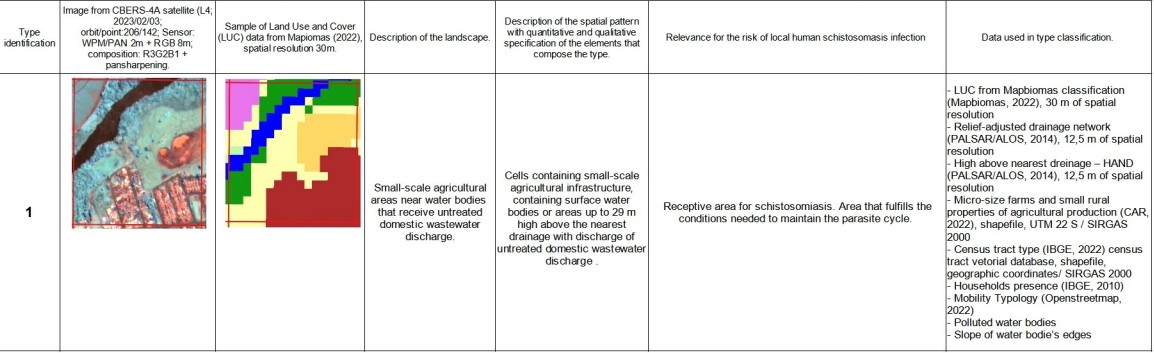

**Fig 4. A Type 1 landscape-unit of potential contact between humans and snails that may be infected.** The landscapes described by this type involve urban and peri-urban areas close to lentic streams with highly favourable habitats for the reproduction of *Biomphalaria sp.*, being small family farms with traditional vegetable cultivation, mainly lettuce, cultivated on flat or slightly sloping, deep, medium-textured soil (sandy-clay or sandy loam), aerated, supplied by a source/spring of water, well drained but with a reasonable water retention capacity, rich in organic matter and with sanitation deficiencies. Satellite image available from http://www.dgi.inpe.br/catalogo/explore. Terms of use available from https://www.gov.br/pt-br/servicos/obter-imagens-de-sensoriamento-remoto-da-terra-geradas-pelo-satelite-cbers-04a.

characterized. The selection of cells at this spatial resolution was guided by empirical testing aimed at assessing how the features of objects present in diverse and heterogeneous databases were portrayed at different cell resolutions. Cells of 100x100, 500x500, and 1000x1000 meters were tested to explore this representation. Considering the limitations in spatial resolution inherent in the databases used in this study, the 500x500-meter resolution emerged as the most suitable in capturing the diversity of structures essential for identifying the landscape units associated with the types described in the typology. With 500x500 metre cells, 73,698 cells were used to cover the entire area of the MP basin.

A set of variables was incorporated into each of the regular grid cells. These variables were derived using different operators which made it possible to integrate information from different data sources with different formats (vector and matrix) and spatial resolutions (Table 2). In this way, it was possible to obtain a disaggregated distribution of the typology's variables, avoiding the problems of spatial non-coincidence between the data representation units. This approach made it possible to create a temporal stability of the units of analysis, aggregated by ad-hoc statistical or computational summarisation measures. This procedure is known as cell filling and, in this work, was carried out using the module in the free and open source Terra-View 5.6.1 GIS software [44, 45].

### Schistosomiasis landscape risk maps

**Map of potential landscapes for Schistosomiasis: Classification and assessment.** Each 500 m X 500 m cell in the MP basin was characterized by the attributes reported in Table 2. We then used a boosted decision tree classification method to classify the cells based on their attributes and assign a landscape type to each cell. The Boosted Decision Tree classifier is an ensemble method that combines multiple decision trees to enhance its predictive accuracy. This method iteratively constructs weak learners (decision trees), focusing on misclassified instances from the preceding iterations. Subsequent trees are adjusted to prioritize these misclassifications, iterating until the model achieves a better overall performance. Boosting, which is based on an adaptive learning method, aims to improve predictive ability by reducing both bias and variance in the model [46, 47]. This method can capture complex non-linear

**Table 2. Landscape grid cell filling operators and the rationale for the inclusion of the variables.**

| Variable | Filling operation | Rationale for inclusion |
|---|---|---|
| Land Use and Land Cover classes | Percentage of each class | Land use and land cover classes allows to measure the observed landscape and the types of human activities (e.g. agriculture). Land use and land cover classes show elements of the physical landscape that can make up the construction of snail habitats. They help to characterise landscapes from a gradient perspective for potential contact situations between humans and snails. |
| Urbanised areas found to be close to areas of the LULC class mosaic of uses | Presence | The mosaic of land use and land cover classes show areas where there is agricultural production on small properties, small areas of secondary vegetation, small areas of clean and dirty pasture, in other words, a class that mixes some types of uses associated with activities carried out by small producers. Generally close to urbanised areas (urban and peri-urban areas). These areas, when present, represent a potential for contact between humans and snails, which may be infected. |
| Situation of the census tract | The situation of the census tract, urban or rural, that was most frequent in each cell. And the situation of the census tract, urban or rural, whose area had the greatest intersection with the area of the cell | Characterisation of the landscape according to their density of occupation, referred to by their urban or rural characterisation. |
| Households | Presence | Presence of households within that landscape unit indicates humans living and circulating in that landscape. |
| Roads according to the Mobility Typology classification | Maximum value | The higher the value assigned to the mobility class within the mobility typology, the more the street is characterized as a pathway for slower and localized movements. |
| Micro-size farms and small rural properties | Presence | Refinement of the agricultural land use class to improve the identification of small-scale agricultural activities. |
| Medium or large production properties | Presence | Refinement of the agricultural land use class to improve the identification of medium and large-scale agricultural activities. |
| Slope at the edge of water bodies | Maximum value | It indicates the slope situation at the edges of water bodies in water collections. |
| Relief-adjusted drainage network | Presence | Indirect characterisation of areas with potential for flooded areas and with conditions for the existence of snails. |
| Water without proper treatment | Presence | Stretches of water in the drainage network that receive untreated domestic effluent. This increases the potential for parasites in these stretches of river present in these landscape units. |
| HAND | Minimum value | The minimum HAND value identifies flooded (or floodable) areas where the conditions exist for finding the snails of interest. |
| Edges of water bodies | Presence | The edges of water bodies in water collections identify places where the snails of interest can be found. |

relationships, patterns and interactions among features, handling different types of data, including numerical and categorical variables. The C5.0 decision tree algorithm [48], implemented in the Geographic Data Mining Analyst (GeoDMA) [49], was used to perform the classification.

Being a supervised method, the decision tree classification requires the use of samples provided by the analyst to train the classifier and assess quality of the classification. The boosting procedure used all the variables present in Table 2 and generated the maximum number of 300 decision trees. The errors of the individual trees during training ranged from 6.9% to 39.0%. The use of boosting, by the voting system, minimizes these tree individual errors by combining the 300 decision tree results trying to reach an error as close to zero as possible. Of the 73,698 cells, 881 cells (1.2%) were evaluated by field work experts, with 563 (63%) used as training samples and 318 (37%) used as testing samples to assess the classification results. The samples were evenly distributed among the ten types of landscape units present in the typology

and spatially distributed along the study area. A confusion matrix was used for the assessment. The assessment of classification models through the confusion matrix approach is a classical one with a consolidated tradition in evaluating model´s predictive performance [50]. This matrix summarizes the model's predictions against actual class labels, detailing the counts of true positive, true negative, false positive, and false negative classifications. It provides a set of metrics for accuracy, precision, recall, and F1-score, which enables a comprehensive understanding of the model's effectiveness.

**Surveillance and control-oriented map for schistosomiasis: Reclassifying landscape units on the map.** The complete map of potential landscapes delineated by the ten types, as outlined in the Typology, was classified. To increase ease of interpretability, a second map was produced to demonstrate the diversity in landscape suitability for three specific scenarios linked to the schistosomiasis at the basin: (i) landscapes with the potential for ongoing transmission, (ii) landscapes deemed receptive, due to their current potential, to initiate or reintroduce active transmission, and (iii) landscapes resistant to the establishment of transmission, exhibiting comparatively lower potential within the current configuration of the basin to initiate or reintroduce active transmission. The set of reclassification rules that was established is depicted in Fig 5.

These new landscape classes are described as follows:

- Potentially Active Landscape for Schistosomiasis: These are landscapes with water collections with habitats favourable to the intermediate host snail, in a transition area between the urban and peri-urban environment, with land use for horticulture, subject to flooding, with the possibility of untreated human waste being disposed of in the collection, and human presence that can develop the infection. Corresponds to types 1 and 2 of the typology.

- Potentially Receptive Landscape to Schistosomiasis: These are landscapes with water collections with favourable habitats for intermediate host snails, in a transition area between urban and peri-urban environments, but also in rural areas, with water collections with riparian vegetation, and without or with low possibility of disposal of untreated human waste. And with low densities of human presence. Corresponds to types 3, 4, 5, 6 and 7 of the typology.

- Potentially Refractory Landscape to Schistosomiasis: These are landscapes with little potential for human transmission of schistosomiasis, either due to the absence of water collections or the absence of human presence or both. Corresponds to types 8, 9 and 10 of the typology.

## Landscape metrics applied to the MP river basin

With the reclassified landscape map, a landscape-level analysis at the MP basin scale can be conducted. The context in which these landscapes exist and their connectivity with neighbouring landscapes can influence the potential risk of infection, the potential for disease reintroduction, and the effectiveness of control and elimination measures for this disease. To assess the spatial arrangement and connectivity of the established landscape categories on the reclassified map, a landscape ecology approach was employed [51]. The three landscape classes (in short)—active, receptive, and refractory—were vectorized and transformed into polygons, maintaining the total area of the MP basin as the scope for analysis. A set of landscape metrics was derived using Fragstats software [52]. The selected metrics for analyzing the spatial arrangement and connectivity of the polygons associated with the landscapes three classes primarily consist of aggregation metrics, measuring the tendency of patch types to be spatially

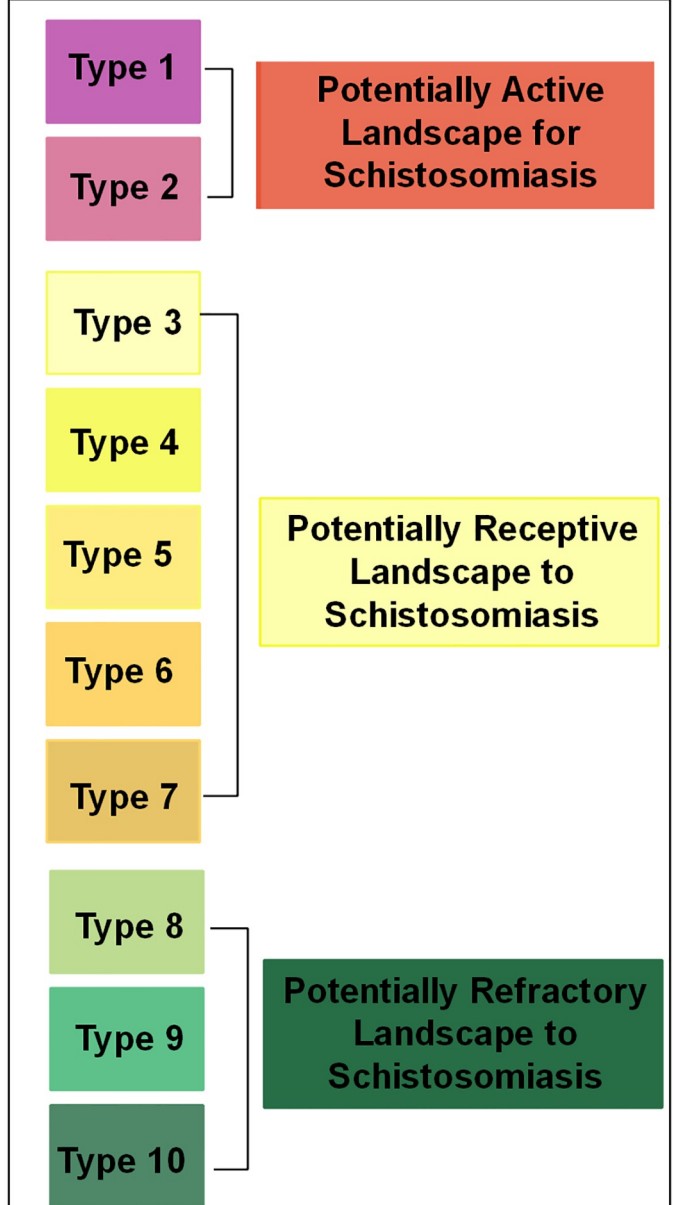

**Fig 5. Reclassification system applied to the schistosomiasis risk landscape map obtained with supervised classification based on machine learning trained on the typology defined for the basin region.**

aggregated. These metrics were used for the individualized analysis of each polygon within each class at two levels: (i) at the class level (C) and (ii) at the landscape level (L) for a comprehensive analysis of the three classes. The selected metrics for this analysis are listed and described in S1 Appendix—Landscape Metrics Description. We utilized the description, when necessary adjusted, from the Fragstats documentation [52].

## Results

First, we present a summary of several outcomes regarding the construction phase of the variables associated with each landscape unit represented by the grid cells covering the entire MP

basin. These variables were employed in the supervised classification process based on boost decision trees.

The methodology employed to correct the property overlaps in the CAR removed 32,706 hectares of overlaps across 23 municipalities within the MP basin, as outlined in S1 Table—CAR Corrected Table. Among these municipalities, Iaras experienced the most significant correction in rural property areas. In this region, the variance between the area declared in the CAR and the corrected property areas amounted to 33.7%.

The outcome of the methodological strategies applied to identify the extent of wetland areas within the MP basin is presented in S2 Fig—Wetlands in the MP Basin. It illustrates drainage and HAND as indicators of freshwater presence, an essential element within environments prone to mollusk fixation. The wetland areas identified, primarily located surrounding the drainage network, cover 43.5% of the entire basin's area.

## The schistosomiasis landscape risk map and its assessment

A boosted decision tree-based supervised classification was applied to assign a specific landscape-unit type to individual grid cells based on their similarity to the provided training samples. The resulting classified map is depicted in Fig 6. Analysis of the map reveals a concentration of Types 1 and 2, more favourable to *S. mansoni* infected snails habitats, in close proximity to watercourses identified as sources of untreated sewage discharge. Type 5, representing agricultural areas near clean water bodies, was the most prevalent type, covering 31.8% of the study area. Types 1 and 2, posing higher potential for the presence of snails potentially infected with *S. mansoni*, accounted for 15.7%. Fig 7 provides a summary of the area coverage within the basin for each type identified in the typology of landscape units associated with schistosomiasis.

## Classification accuracy assessment

A confusion matrix (Fig 8) was generated to assess the classification accuracy. Each evaluation sample's assigned value after classification was compared to the value provided by the analyst. The overall classification accuracy stood at 0.925, indicating that 92.5% of the 318 cells designated for evaluation were correctly classified.

The assessment also involved a database containing snail data gathered from 2015 to 2018 in five municipalities within the Middle Paranapanema watershed [30]. Sampling encompassed 654 specific locations across 114 water bodies in the municipalities of Ourinhos, Ipaussu, Ribeirão do Sul, Assis, and Chavantes. Within these sampling points, 3151 snails belonging to at least one of the three intermediate host species of schistosomiasis were identified at 43 locations. Fig 9 displays a map featuring the classified landscape types alongside the collected *Biomphalaria* snails in the field. The assessment factored in the quantity of mollusks and the locations of collection points, each recording at least one individual from the species of intermediate host for schistosomiasis.

The majority of the snails, constituting 74.6% of the total count, were collected in areas classified as Types 1 and 2. *B. glabrata* stands as the most susceptible neotropical snail host to *S. mansoni* within the MP basin region. This particular snail species has historically been associated with the incidences of schistosomiasis among the human population living in the basin. Taking into account eight collection points that exclusively exhibited *B. glabrata*, the distribution was as follows: Type 1 with 1 collection point comprising 100 snails; Type 2 with 2 collection points totaling 258 snails; Type 6 with 1 collection point accounting for 46 snails; Type 7 with 3 collection points amounting to 86 snails and a Type 9 landscape-unit pattern with 1 collection point with 12 snails. Of the 502 *B. glabrata* collected, 71.3% were found in landscape

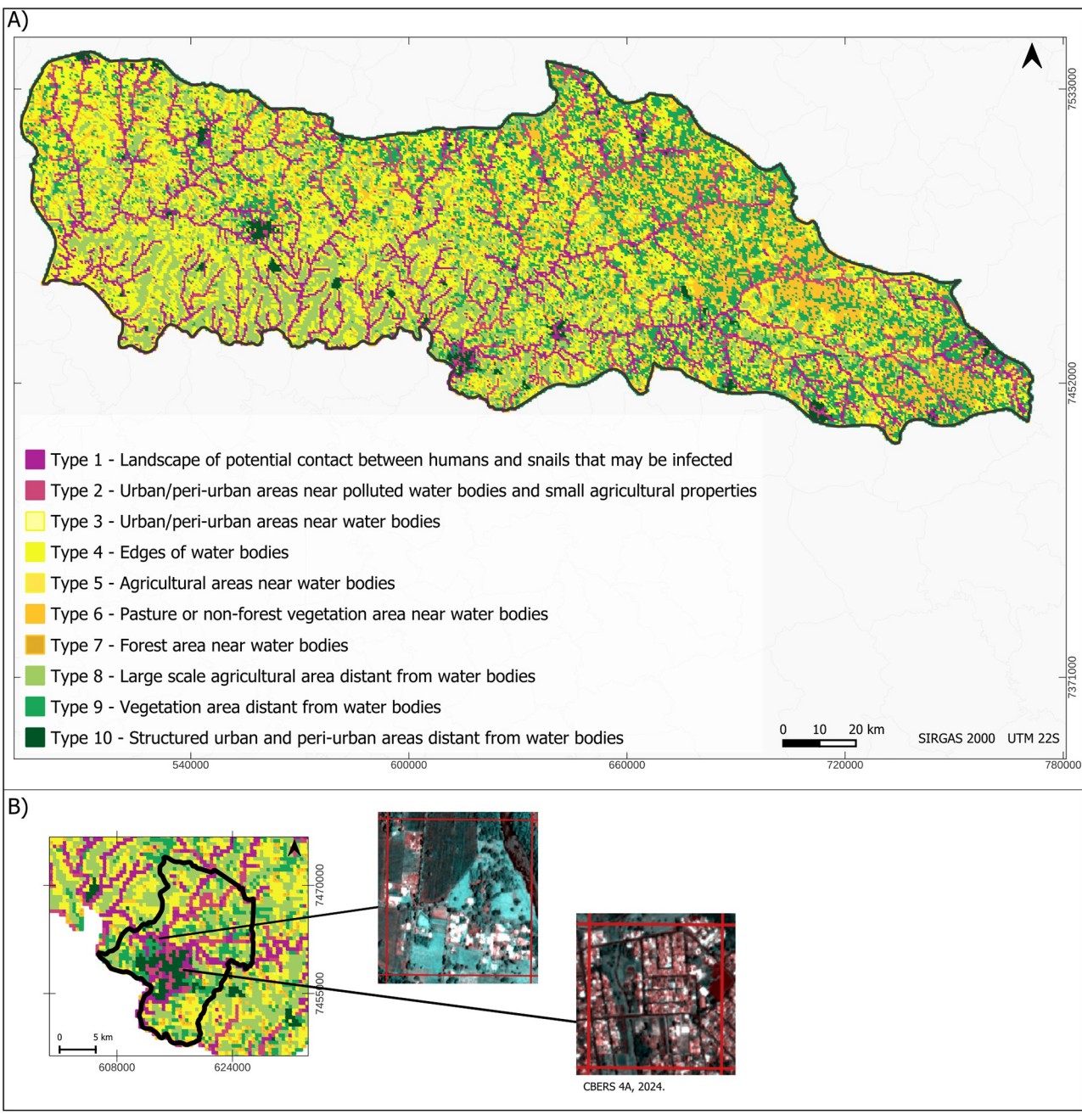

**Fig 6. Schistosomiasis landscape risk map.** (A) Mapping depicting the 10 types of landscape units within the MP basin utilizing the proposed typology, derived through supervised classification employing the boost decision tree method. (B) Emphasized section showcasing images associated with the municipality of Ourinhos. Municipality border shape available from https://geoftp.ibge.gov.br/organizacao_do_territorio/malhas_territoriais/ malhas_municipais/municipio_2022/Brasil/BR/BR_Municipios_2022.zip. Terms of use available from https://biblioteca.ibge.gov.br/visualizacao/livros/ liv101998.pdf. Satellite image available from http://www.dgi.inpe.br/catalogo/explore. Terms of use available from https://www.gov.br/pt-br/servicos/ obter-imagens-de-sensoriamento-remoto-da-terra-geradas-pelo-satelite-cbers-04a. MP border shape available from https://datageo.ambiente.sp.gov.br/ geoserver/datageo/LimiteUGRHI/wfs?version=1.0.0&request=GetFeature&outputFormat=SHAPE-ZIP&typeName=LimiteUGRHI. License information available from https://datageo.ambiente.sp.gov.br/sobre.

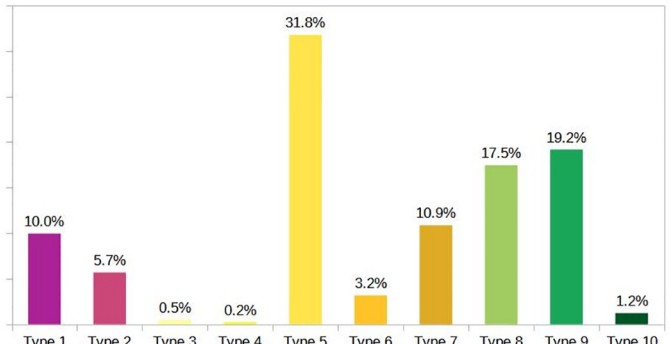

**Fig 7. Percentage of area within the basin for each landscape-unit type.**

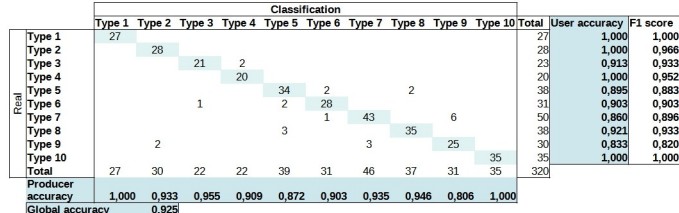

**Fig 8. Confusion matrix depicting the assessment of classification accuracy.**

units classified as having a high potential for contact between hosts, potentially infected, and humans—specifically, Types 1 and 2. Additionally, 26.3% of the snails were located in landscape units identified as having a high potential for snail presence, albeit with a lower probability of encountering infected snails. Moreover, 2.4% of the snails, found solely at one collection point, were situated in a landscape unit classified as highly unsuitable for snail presence. In S3 Fig—*B. glabrata* in Landscape unit Type 9, we present some potential factors associated with this classification error.

## Surveillance and control-oriented schistosomiasis landscape risk map

Fig 10 displays the reclassified map of the 10 landscape units identified through the supervised classification process, condensed into just three reclassified landscape units. This reorganization of landscape units provides a cartographic simplification intended for use by schistosomiasis surveillance and control services in the basin region. Three major landscapes of significance for surveillance and control actions at regional and local scales were established as: (i) Potentially Active Landscape, (ii) Potentially Receptive Landscape, and (iii) Potentially Refractory Landscape. In short, active, receptive and refractory landscapes for shistosomiasis in the basin.

## A panel for communicating the results to the surveillance and control teams in the basin area

An online open-access interactive dashboard has been developed for the schistosomiasis surveillance and control teams in the basin area. The panel is currently in its preliminary version and is under discussion with both management and field teams. The current version of the

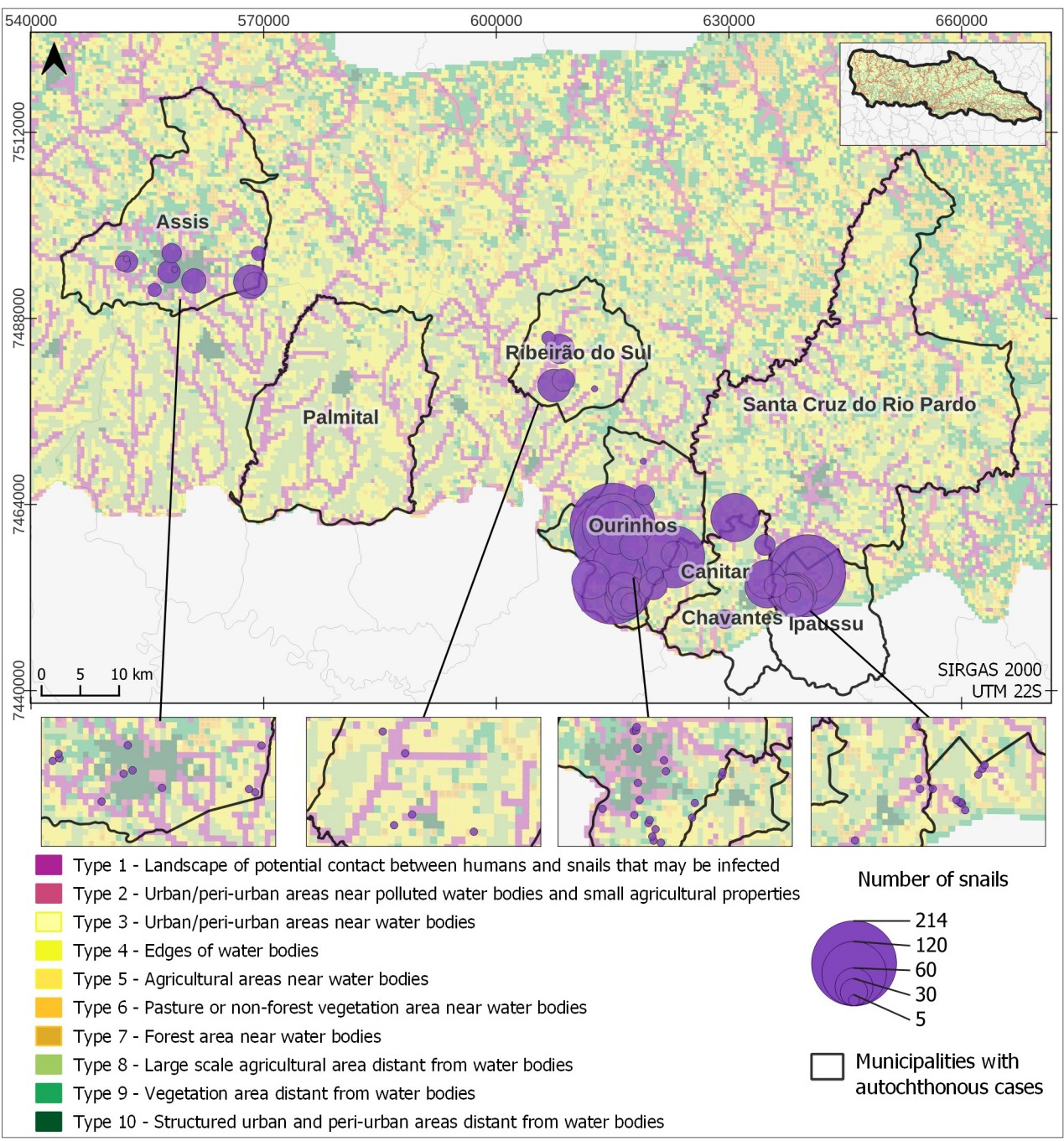

**Fig 9. Map of classified landscape types and the collected *Biomphalaria* snails in the field.** Municipality border shape available from https://geoftp. ibge.gov.br/organizacao_do_territorio/malhas_territoriais/malhas_municipais/municipio_2022/Brasil/BR/BR_Municipios_2022.zip. Terms of use available from https://biblioteca.ibge.gov.br/visualizacao/livros/liv101998.pdf. MP border shape available from https://datageo.ambiente.sp.gov.br/ geoserver/datageo/LimiteUGRHI/wfs?version=1.0.0&request=GetFeature&outputFormat=SHAPE-ZIP&typeName=LimiteUGRHI. License information available from https://datageo.ambiente.sp.gov.br/sobre.

panel presents maps resulting from the complete classification of the 10 types of landscapes and the reclassified map featuring potentially active, receptive, and refractory landscapes. The dashboard comprises four content pages: the first page displays the results from supervised classification of the 10 landscape types associated with schistosomiasis; the second page

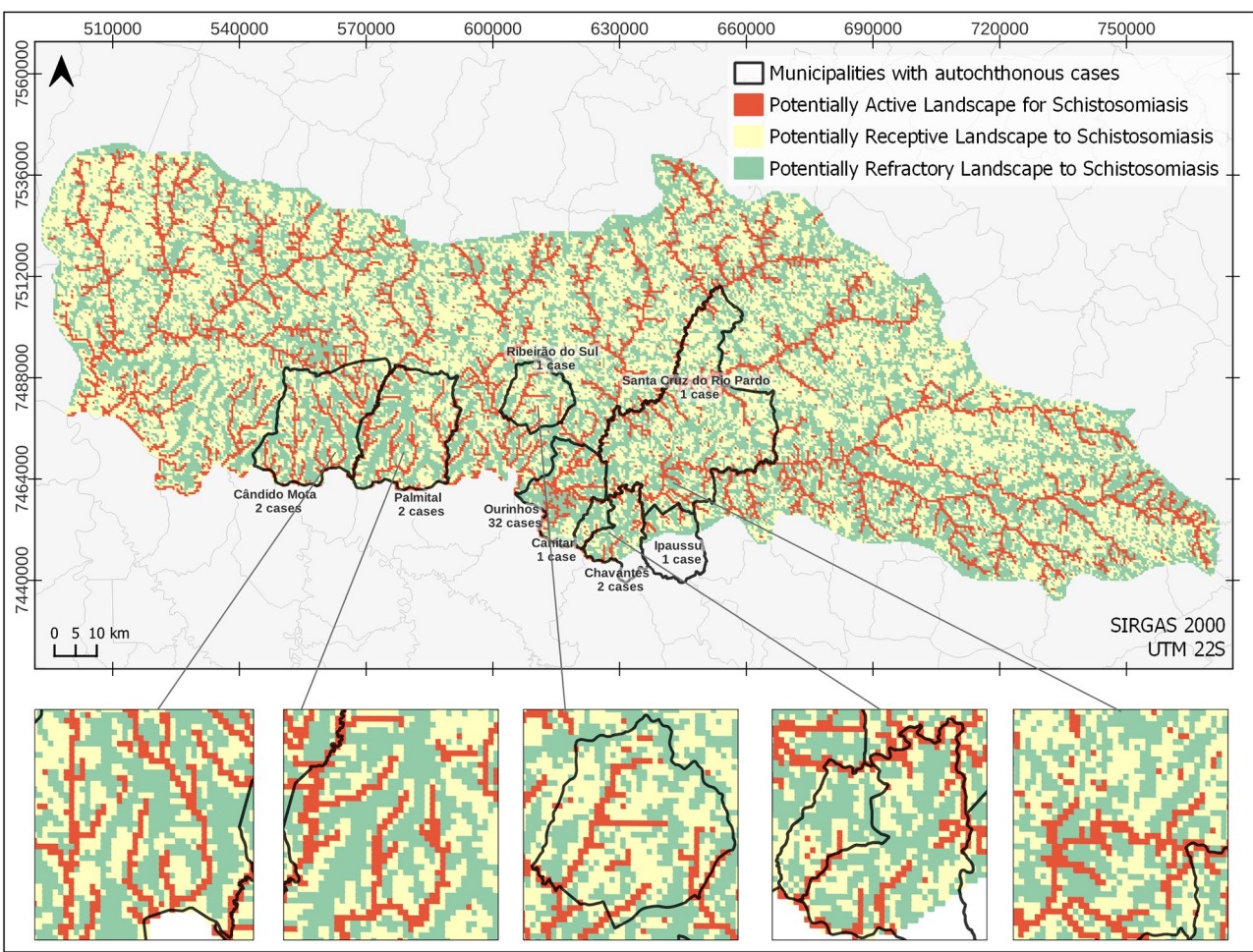

**Fig 10. Map depicting landscapes posing schistosomiasis risks, featuring three specific classes—Active, receptive and refractory—For formulating monitoring and controlling strategies aimed at regional and local programs.** Municipalities that reported autochthonous cases of schistosomiasis from 2006 to 2022 were highlighted. Municipality border shape available from https://geoftp.ibge.gov.br/organizacao_do_territorio/malhas_territoriais/ malhas_municipais/municipio_2022/Brasil/BR/BR_Municipios_2022.zip. Terms of use available from https://biblioteca.ibge.gov.br/visualizacao/livros/ liv101998.pdf. MP border shape available from https://datageo.ambiente.sp.gov.br/geoserver/datageo/LimiteUGRHI/wfs?version=1.0.0&request= GetFeature&outputFormat=SHAPE-ZIP&typeName=LimiteUGRHI. License information available from https://datageo.ambiente.sp.gov.br/sobre.

exhibits the reclassification outcomes with aggregated data forming landscape classes potentially active, receptive, or refractory to schistosomiasis; the third page accommodates the document outlining the typology of landscape patterns associated with schistosomiasis in the basin area and its description, while the fourth page shows the confusion matrix generated with evaluation samples. This version is available only in Portuguese as its objective is direct interaction with the services. This dashboard version can be accessed at (https://anrio0-vivian-silva. shinyapps.io/esquistossomose3/). For the construction of this Dashboard in its initial version, the R programming language and packages like Leaflet, Plotly, and Shiny were utilized as the foundation for the code development, available at https://github.com/vafds/esquistossomose. git. This version is available only in Portuguese as its aim is direct engagement with the services (Fig 11).

Landscape metrics were examined at two levels on a basin scale: class (C) and landscape (L). At this broader scale the analysis was carried out using mainly aggregation measures for

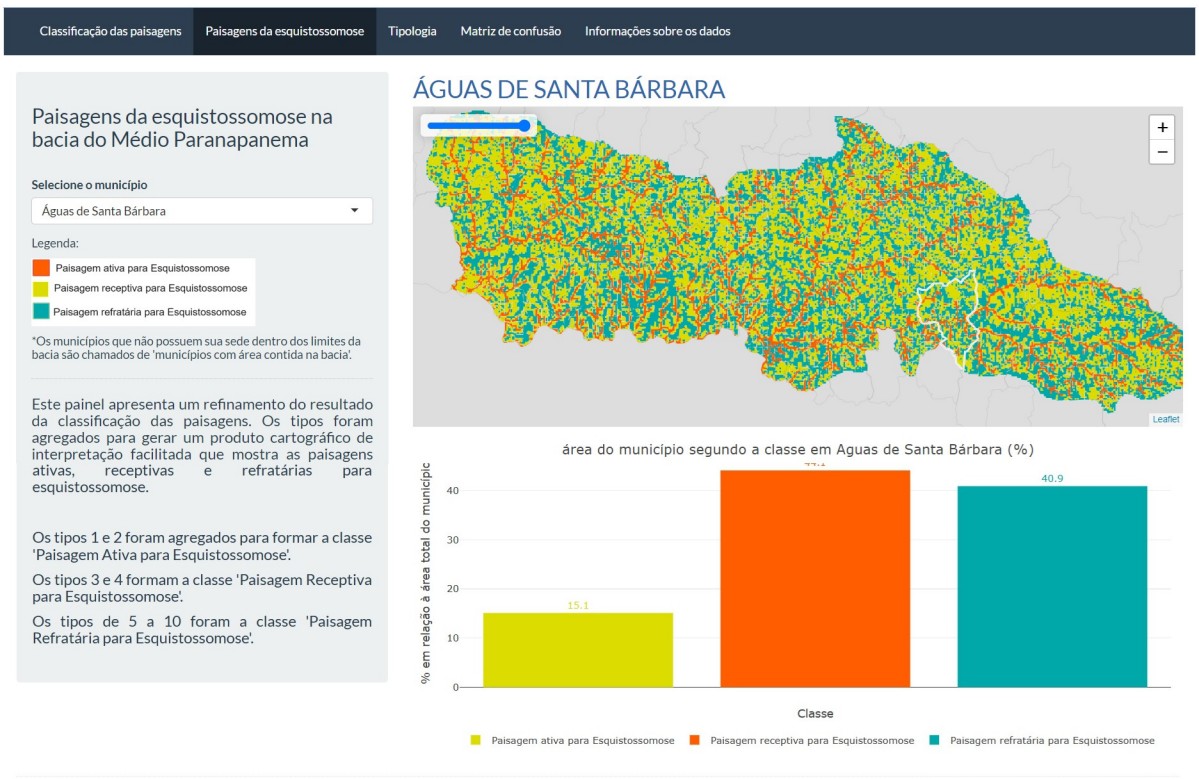

**Fig 11. Second content page of the schistosomiasis dashboard.** It shows the complete map of potentially active, receptive, and refractory landscapes in the interactive panel (Dashboard). The municipality being observed is Águas de Santa Bárbara. At the top, links to the other three content pages are seen. Municipality border shape available from https://geoftp.ibge.gov.br/organizacao_do_territorio/malhas_territoriais/malhas_municipais/municipio_2022/Brasil/BR/BR_Municipios_2022.zip. Terms of use available from https://biblioteca.ibge.gov.br/visualizacao/livros/liv101998.pdf. MP border shape available from https://datageo.ambiente.sp.gov.br/geoserver/datageo/LimiteUGRHI/wfs?version=1.0.0&request=GetFeature&outputFormat=SHAPE-ZIP&typeName=LimiteUGRHI. License information available from https://datageo.ambiente.sp.gov.br/sobre.

the schistosomiasis landscape risk classes. Table 3 shows these metrics results that are fully analysed in the discussion session.

## Discussion

Schistosomiasis affects populations within their areas of life and experiences where, generally, the implementation of Water, Sanitation, and Hygiene (WASH) strategies is either non-

**Table 3. Metrics extracted from the reclassified map showing the three aggregated landscape types in the Middle Paranapanema river basin: Potentially active, receptive and refractory landscapes patterns.** At the class level (C): Aggregation Index (AI), Proportion of Landscape (PLAND), Percentage of Like Adjacencies (PLADJ) and a Fragmentation Index (SPLIT) and at the landscape level (L): Mean Euclidean Nearest Neighbor Distance (ENN_MN).

| (L) Landscape Level | | | |
|---|---|---|---|
| ENN_MN | 688 | | |
| (L) Landscape Level | (L) Potentially Active | (L) Potentially Receptive | (L) Potentially Refractory |
| PLAND | 38.1 | 51.8 | 10.1 |
| AI | 95.8 | 96.6 | 94.7 |
| PLADJ | 95.7 | 96.5 | 94.6 |
| SPLIT | 122.4 | 74.8 | 7194.0 |

existent or inadequately covered. This circumstance promotes varying degrees of contact and exposure between *S. mansoni*-infected intermediate hosts and humans. The landscapes shared by snails, the parasitic trematode, and humans are increasingly interwoven. These landscapes remain in constant transformation, with dynamics varying in speed. Changes in the physical landscape, alterations in native coverages, and modifications in land use, stemming from decisions associated with urban development models and ways of life imposed by societal choices, further drive these transformations. Associated with these socio-environmental dynamics are new dynamics resulting from the local implications of global-scale climate changes, such as an increase in the frequency and intensity of extreme phenomena like precipitation volume in the basin. These new elements add to the existing ones, shaping and reshaping the territories inhabited by humans and snails, thereby creating an integrated landscape, a physical-social complex that exhibits spatial and temporal arrangements leading to, or potentially leading to, the emergence, re-emergence, consolidation, or elimination of schistosomiasis. In summary, human collective actions, further compounded by the new situations arising from climate change reflected in local territories, can disrupt the local/regional landscape configurations and affect the establishment, maintenance and or elimination of the schistosomiasis transmission cycle at the basin scale.

This study aimed to examine this scale, that of the transformed landscape, as a pathogenic complex of schistosomiasis in the MP basin region. It highlights for the services the landscapes within a temporal scope, establishing the methodological foundations for monitoring their transformation. In this sense, the methodology proposed and tested in this study for the MP basin establishes a new technical-instrumental possibility for the services associated with the surveillance and control of schistosomiasis in the area. In the context of low endemicity, the map emerges as an important tool to support localized action strategies in hot spots to eliminate schistosomiasis and avoid its resurgence. Additionally, the map can also guide public efforts to expand coverage of basic sanitation in areas requiring such investments, specifically in potentially active landscapes for schistosomiasis.

Taking these issues into consideration, the decision to construct a set of variables incorporating local data and knowledge to define the landscape types associated with schistosomiasis in the basin proved to be essential. The inclusion of a simplified model for detecting polluted water among the variables in this study was crucial for a classification outcome more aligned with the reality of domestic effluent discharge in the study area. Concerning the drainage network incorporated in the classification, adjustments regarding the relief forms aimed to prevent underestimation or overestimation of the hydrological configuration, avoiding major issues with the pollution model of specific sections within the network. The 12.5-meter spatial resolution of the PALSAR sensor aboard the ALOS satellite allowed for better recognition of terrain relief nuances compared to other freely used Digital Terrain Models (DTMs) commonly employed, such as the Shuttle Radar Topography Mission (SRTM), which possess lower spatial resolution [35, 53].

The HAND from the drainage network adjusted to the terrain relief was utilized to establish potential infection areas within the aquatic environment. Unlike the area of influence or buffer, which would indicate the horizontal distance between the drainage and a specific point, HAND accounts for variations in terrain altitude and includes only areas susceptible to flooding by adjacent watercourses. The cutoff point for the HAND metric disregarded areas with values exceeding the maximum height found through overlaying data on intermediate host snail collection in the region. This extrapolated data, applicable to the entire study area, is available for five municipalities within the Middle Paranapanema basin, leading to a non-individualized cutoff point for each municipality.

The correction of overlaps in Rural Environmental Registry (CAR) properties revealed that 2.83% of the total area of rural properties in the Middle Paranapanema basin exhibited rectified overlaps across 23 municipalities. Predominantly rural or peri-urban areas cover the major portion of the study area, despite most census tracts in the Middle Paranapanema basin being officially classified as "urban". This classification arises from the fact that the average size of rural sectors is larger than that of other classes, characterized by low household density.

An important limitation emerged in the land use and land cover map derived from satellite images, using the MapBiomas product. Because it is based on Landsat satellite data (with a 30-meter spatial resolution), the agricultural-related classes cannot discern more specific horticultural areas, such as vegetable and greens production belts, or even small backyard gardens in peri-urban areas. However, MapBiomas is currently the finest-resolution land-use classification data source that is publicly available for this region. By associating it with CAR data, there was a refinement, although not as desired, of these agricultural and livestock production areas. Undoubtedly, mapping with finner spatial resolution will yield a land use-related variable better suited for the classification process.

And lastly, but not less importantly, the generated maps offer us the significant opportunity to analyse the landscape's situation in the basin, considering the spatial distribution and connectivity conditions related to classes potentially active, receptive, and refractory to schistosomiasis in the basin area.

Using the selected landscape metrics, a simple measure of the association between spatial patterns of landscapes and the processes involved in the transmission cycle is the area proportion, PLAND(C). This index was calculated at 10%, 52%, and 38% for landscapes classified as active, receptive, and refractory, respectively. Landscape metrics estimated at the class level show that the polygons constituting these landscapes have a high level of aggregation. Landscapes classified as potentially active, despite occupying a smaller area proportion in the basin (10%) compared to the other classes, exhibit high aggregation indices, approaching 100 (maximum aggregation value). This can be observed in the AI(C) and PLADJ(C) indices presented in Table 3. At the landscape level, the average Euclidean distance to the nearest polygon, ENN_MN (L), considering all classes, was estimated at 688 meters. This result suggests that the polygons representing different landscapes are relatively close to each other, potentially connecting based on changes in climatic, socio-environmental, and political scenarios that govern the basin territories.

The level of landscape fragmentation is an important element to complement the analysis. We consider that the more subdivided the polygon classes are within a landscape, the higher the degree of isolation of that landscape in relation to risk classes. Isolation, concerning the risk of disease transmission, generally serves as a protective factor. For the landscape fragmentation analysis, we used the SPLIT(C) metric, which assessed each of the 3 classes. As a result, we obtained the highest value (7,194) for the potentially active class, indicating a high level of fragmentation and a potential isolation of some polygons composing this category. Although most of the area in this class is aggregated in larger polygons, numerous small, non-contiguous polygons are noticeable close to these aggregates (Fig 10). In summary, the spatial characteristics analysed reveal that polygons representing potentially active landscapes are largely grouped but demonstrate a high level of fragmentation and numerous isolated small patches. This helps to explain, partially, the low endemicity we found in the basin region. In the connectivity analysis using the ENN_MN distance index, we observed that polygons from the 3 landscape types are close to each other, indicating that the connection of the active class with the others (receptive and refractory) due to the spatial characteristics highlighted in this analysis could be facilitated in case of changes in the landscape configuration at the basin scale. This could be influenced by a set of endogenous and exogenous factors altering the land use and

land cover mosaics and coverage situations for WASH-related services, particularly crucially important, the local and regional surveillance and control services that were not included in this modelling. Improvements in WASH-related conditions from 2000 to 2010, coupled with institutional strengthening of surveillance and control during the same period, are additional components that contribute to maintaining the observed low endemicity situation.

Extreme weather events (such as increased rainfall in short periods leading to flooding) or changes in surveillance actions can modify landscapes and transmission conditions in areas, connecting refractory and receptive landscapes to active landscapes, thereby altering the current scenario of schistosomiasis transmission control in the basin.

## Conclusion

This work offers an effective basis for an integrative view for schistosomiasis risk mapping at the landscape scale for conducting research and promoting the reproducibility of methodologies. Our selection of free, open and reliable data with extensive national coverage enables the presented methodology to be replicated in other basins. A crucial element, besides the data, is the participation of local surveillance teams and the involvement of these teams and regional experts in defining the schistosomiasis typologies and the strategies for field visits in order to refine it.

The resulting classified landscape-unit types were rearranged to produce a map that expresses the gradient of possibility of local transmission of schistosomiasis. Landscape metrics were extracted from the final cartographic product, and their interpretation helped to understand the current scenario of low endemicity of schistosomiasis in the study area. The findings suggest that the areas with the highest risk of establishing the schistosomiasis transmission cycle are more isolated and disconnected in the Middle Paranapanema river basin.

Our study has introduced an innovative methodology that leverages open Earth Observation data, open geotechnologies and open machine learning based supervised classification methods, that allows for introducing different levels of local and regional knowledge into the flow of technical procedures. Our findings amplify and extend existing mapping approaches by integrating environmental, epidemiological and socio-economic data in search of elucidating the intricate interplay between human society and nature, which together have influence on the transmission cycle of schistosomiasis. However, the analysis presented here is not exhaustive. For example, the intra-urban aspect could be further explored through broader-scale mapping initiatives to depict the disparities within urban areas. The spatial insights derived from this study serve as a valuable resource for implementing interventions aimed at reducing the population requiring treatment for neglected tropical diseases, aligning with the objectives of the World Health Organization and national schistosomiasis control and elimination programs.

However, some limitations of our study need to be addressed. Initially, the geographic distribution of *B. glabrata*, *B. tenagophila*, and *B. straminea* in the MP is unequal and scattered. Actually, *B. glabrata* habitats have been restricted to a few sites over time, which contrasts with the distributions of both of the other two species. Furthermore, the three snail hosts used in this study were considered equally susceptible to *S. mansoni*. However, important investigations have clearly established intra- and interspecific differences in neotropical *Biomphalaria/ S. mansoni* [54–57]. Since *B. glabrata* is the most susceptible neotropical host to *S. mansoni*, this species has historically been connected to human occurrences of schistosomiasis in the MP basin. But how can we explain the decrease in schistosomiasis in the MP considering the fact that *B. tenagophila* is still abundant in water collections? Murray and Daszak [58] propose two ecological theories to explain the dynamics of diseases: 1-disturbance and 2-pathogen

pooling. So, landscape disturbance in the MP basin might has favoured colonization of a more diverse freshwater snail species, including *B. occidentalis* and *B. peregrina*. This recent pattern of freshwater colonization would possibly, in some way, act as a buffer for local schistosomiasis transmission. Can schistosomiasis strongly reemerge in the MP basin region? Considering the climate changes and its impact on floods; the rapid regional urbanisation processes, which has not been accompanied by an expansion in the coverage and quality of services associated with WASH strategies; the weakening of disease surveillance and control institutions by the state of São Paulo, through the dissolution of SUCEN-Superintendência de Controle de Endemias (Superintendence of Endemic Disease Control, established on April 17, 1970) by Decree no. 66.664 on April 14th, 2022, which addresses the effective dissolution of SUCEN, and the MP extensive vulnerables areas for flooding, indicate that the risk for infection is still a hypothesis for this region.

## Supporting information

**S1 Fig. Typology of landscape patterns associated with schistosomiasis in the Middle Paranapanema watershed.** The figure shows 10 types of landscape units associated with schistosomiasis in the MP basin. To describe and characterise these types, we used (i) an image from the CBERS-4A satellite (L4; 2023/02/03; orbit/point:206/142; Sensor WPM/PAN 2m + RGB 8m; composition: R3G2B1 + pansharpening); (ii) a selected sample of land use and land cover data from MapBiomas (2022); (iii) a general description of that landscape associated with the type; (iv) Description of the spatial patterns observed in the landscape with quantitative and qualitative characterisation for the elements that make up that landscape; (v) a description of the relevance of that landscape in the composition of the local risk for schistosomiasis; and (vi) the data used to generate the variables/indexes that took part in the supervised classification process based on a machine learning approach. Satellite image available from http://www.dgi. inpe.br/catalogo/explore. Terms of use available from https://www.gov.br/pt-br/servicos/ obter-imagens-de-sensoriamento-remoto-da-terra-geradas-pelo-satelite-cbers-04a. (TIF)

**S2 Fig. Wetlands in the MP basin.** A) The map shows the outcome of the methodological strategies applied to identify the extent of wetland areas within the MP basin. It illustrates drainage network and high above nearest drainage (HAND) as indicators of freshwater presence. The wetland areas identified, primarily located surrounding the drainage network, cover 43.5% of the entire basin's area. B) The satellite image shows the difference between the two drainage network in different areas. The ultimate drainage network was constituted by the amalgamation of these two components, with its configuration influenced by the topographical features. Municipality border shape available from https://geoftp.ibge.gov.br/organizacao_ do_territorio/malhas_territoriais/malhas_municipais/municipio_2022/Brasil/BR/BR_ Municipios_2022.zip. Terms of use available from https://biblioteca.ibge.gov.br/visualizacao/ livros/liv101998.pdf. Satellite image available from http://www.dgi.inpe.br/catalogo/explore. Terms of use available from https://www.gov.br/pt-br/servicos/obter-imagens-de-sensoriamento-remoto-da-terra-geradas-pelo-satelite-cbers-04a. MP border shape available from https://datageo.ambiente.sp.gov.br/geoserver/datageo/LimiteUGRHI/wfs?version=1.0. 0&request=GetFeature&outputFormat=SHAPE-ZIP&typeName=LimiteUGRHI. License information available from https://datageo.ambiente.sp.gov.br/sobre. (TIF)

**S3 Fig. *B. glabrata* in landscape unit Type 9.** The figure shows the two *B. glabrata* in landscape unit Type 9 collection points near water bodies in cells with other land use and land

cover. The classification rules, exemplified by parameters like the proportion of cell area occupied by forest vegetation or urbanized features, served to designate these areas as distinct from other landscape types. Satellite image available from http://www.dgi.inpe.br/catalogo/explore. Terms of use available from https://www.gov.br/pt-br/servicos/obter-imagens-de-sensoriamento-remoto-da-terra-geradas-pelo-satelite-cbers-04a.
(TIF)

**S1 Appendix. Landscape metrics description.** Description of the metrics employed in the analysis of individual polygons belonging to the three distinct classes at two levels: (i) at the class level (C) and (ii) at the landscape level (L).
(PDF)

**S1 Table. CAR corrected table.** The table presents the 23 municipalities within the MP basin wherein the methodology applied for rectifying overlaps in the Cadastro Ambiental Rural (CAR) corrected 32,706 hectares of overlap. Each municipality is accompanied by information detailing the declared area in the CAR, the corrected area subsequent to rectification, and the corresponding percentage disparity between these two values.
(PDF)

**S1 Data. Cases of schistosomiasis in Middle Paranapanema river basin municpalities, *Biomphalaria* snails collected by Palasio et al. (2019) [31], area of each landscape-unit type by municipality, area of each class by municipality, wetland area in MP river basin.**
(XLSX)

# Acknowledgments

The authors thank the technical support of the fieldwork team of the Center for Health Control of the São Paulo State Health Authority based at Ourinhos assisting in all fieldwork activities and facilitating access to the freshwater bodies and snails collection.

# Author Contributions

**Conceptualization:** Vivian Alessandra Ferreira da Silva, Milton Kampel, Maria Isabel Sobral Escada, Antônio Miguel Vieira Monteiro.

**Data curation:** Vivian Alessandra Ferreira da Silva, Rafael Silva dos Anjos, Raquel Gardini Sanches Palasio, Roseli Tuan.

**Formal analysis:** Vivian Alessandra Ferreira da Silva.

**Investigation:** Vivian Alessandra Ferreira da Silva, Milton Kampel, Rafael Silva dos Anjos, Raquel Gardini Sanches Palasio, Maria Isabel Sobral Escada, Roseli Tuan, Alyson Singleton, Caroline Kate Glidden, Andrew Chamberlin, Giulio Alessandro De Leo, Adriano Pinter dos Santos, Antônio Miguel Vieira Monteiro.

**Methodology:** Vivian Alessandra Ferreira da Silva, Milton Kampel, Maria Isabel Sobral Escada, Roseli Tuan, Antônio Miguel Vieira Monteiro.

**Validation:** Vivian Alessandra Ferreira da Silva.

**Visualization:** Vivian Alessandra Ferreira da Silva.

**Writing – original draft:** Vivian Alessandra Ferreira da Silva, Antônio Miguel Vieira Monteiro.

**Writing – review & editing:** Vivian Alessandra Ferreira da Silva, Milton Kampel, Rafael Silva dos Anjos, Raquel Gardini Sanches Palasio, Maria Isabel Sobral Escada, Roseli Tuan, Alyson Singleton, Caroline Kate Glidden, Andrew Chamberlin, Giulio Alessandro De Leo, Adriano Pinter dos Santos, Antônio Miguel Vieira Monteiro.

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
