## [Decision Letter · Decision Letter 0]

21 May 2024

Dear Silva,

Thank you very much for submitting your manuscript "Mapping schistosomiasis risk landscapes and implications for disease control: a case study for low endemic areas in the Middle Paranapanema river basin, São Paulo, Brazil" for consideration at PLOS Neglected Tropical Diseases. As with all papers reviewed by the journal, your manuscript was reviewed by members of the editorial board and by several independent reviewers. In light of the reviews (below this email), we would like to invite the resubmission of a significantly-revised version that takes into account the reviewers' comments. 

We cannot make any decision about publication until we have seen the revised manuscript and your response to the reviewers' comments. Your revised manuscript is also likely to be sent to reviewers for further evaluation.

Sincerely,

Paul O. Mireji, PhD

Section Editor

Paul Mireji

Section Editor

Reviewer's Responses to Questions

**Key Review Criteria Required for Acceptance?**

**Methods**

-Are the objectives of the study clearly articulated with a clear testable hypothesis stated?

-Is the study design appropriate to address the stated objectives?

-Is the population clearly described and appropriate for the hypothesis being tested?

-Is the sample size sufficient to ensure adequate power to address the hypothesis being tested?

-Were correct statistical analysis used to support conclusions?

-Are there concerns about ethical or regulatory requirements being met?

Reviewer #1: -Are the objectives of the study clearly articulated with a clear testable hypothesis stated? Yes

 -Is the study design appropriate to address the stated objectives? Yes

 -Is the population clearly described and appropriate for the hypothesis being tested? Yes

 -Is the sample size sufficient to ensure adequate power to address the hypothesis being tested? Yes

 -Were correct statistical analysis used to support conclusions? Yes

 -Are there concerns about ethical or regulatory requirements being met? Yes

Reviewer #2: The research objectives are clearly defined, and the design seems appropriate for the goals. However, the excerpt does not provide enough information to fully assess the sample size's adequacy or ethical considerations. The statistical analysis appears to be correctly applied based on the information given.

Reviewer #3: The objectives of the study are clearly articulated

The study design is appropriate to address the stated objectives

The population is clearly stated

The sample size is adequate

The correct statistical analysis were used

The authors should inculde an ethical statement for the study

**Results**

-Does the analysis presented match the analysis plan?

-Are the results clearly and completely presented?

-Are the figures (Tables, Images) of sufficient quality for clarity?

Reviewer #1: -Does the analysis presented match the analysis plan? Yes

-Are the results clearly and completely presented? Yes

-Are the figures (Tables, Images) of sufficient quality for clarity? Yes

Reviewer #2: In summary, the results section of the manuscript seems to be well-executed, with a clear and thorough presentation of findings that match the pre-established analysis plan. The approach to evaluating and discussing the results is systematic and provides a solid foundation for the study's conclusions.

Reviewer #3: The analysis are excellent, results clearly presented with all the figures and tables being of high quality

**Conclusions**

-Are the conclusions supported by the data presented?

-Are the limitations of analysis clearly described?

-Do the authors discuss how these data can be helpful to advance our understanding of the topic under study?

-Is public health relevance addressed?

Reviewer #1: -Are the conclusions supported by the data presented? Yes

-Are the limitations of analysis clearly described? Yes

-Do the authors discuss how these data can be helpful to advance our understanding of the topic under study? Yes

-Is public health relevance addressed? Yes

Reviewer #2: The conclusion section effectively summarizes the study's findings, is supported by the data, and acknowledges its limitations. The authors also successfully link their research to its potential impact on public health policy and practice, particularly in the context of disease surveillance and control. This approach not only strengthens the study's academic contribution but also its practical utility for public health initiatives.

Reviewer #3: The study conclusions are fully and adequately supported by the data presented

The study limitations are clearly described

The public health relevance is addressed

The authors have discussed conclusively the importance of their data to the advancement of the topic under study

**Editorial and Data Presentation Modifications?**

Reviewer #1: My comments are shown in the text of the attached MS.

Reviewer #2: 1. Lines 142-167: The article utilizes multiple data sources, including remote sensing imagery, census data, and socio-economic data. This is a strength of the study as it allows for a multifaceted analysis of the issue. However, is it possible that differences in data quality and resolution could affect the accuracy of the results?

2. Lines 334-341: The use of Boosted Decision Tree as a classification method in the study is a reasonable choice because it can handle complex nonlinear relationships. It may be worth considering providing additional details on the rationale for model selection and the parameter tuning process to enhance transparency in the methods section.

3. Has the study considered the generalizability of the model to other regions or under different conditions? Validating the applicability of the model in new areas is important.

4. Lines 608-610: The conclusion mentions the potential impact of climate change on the spread of schistosomiasis. How do these changes specifically affect risk maps, and what are the possible adaptation strategies?

5. Lines 612-613: The study provides detailed risk maps, which are highly useful for disease control and intervention strategies. Have considerations been made on how these maps specifically impact policy-making and fieldwork?

6. Lines 622-626: The transmission of schistosomiasis is a dynamic process influenced by multiple factors. Has there been consideration for regularly updating risk maps to reflect changes in environmental and socio-economic conditions?

Reviewer #3: Author summary- Line 3, put citation in to brackets (Sambon , 1907)

Methodology: study area: Line 138- repetition of the word ‘from’

Methodology: Physical landscape data: Lines 173-175: Was potential agricultural areas which could be high potential areas for human parasite interactions captured under Open Street Map (OSM)?

Include an ethical statement for the study-Did the study obtain ethical approval?

**Summary and General Comments**

Reviewer #1: The theme of the manuscript is undoubtedly of great interest to public health as schistosomiasis is currently among the most important neglected diseases in the tropics. In the specific case of Brazil, the focus of this Manuscript, although the last prevalence survey carried out indicated a significant decrease in the prevalence of schistosomiasis, there are still areas with high prevalence, in addition to the persistence of those with low prevalence and some focal areas of schistosomiasis mansoni. In recent years, there is a consensus that the control and possible elimination of schistosomiasis requires the application of several integrated measures, as it is well presented in this Manuscript. The authors present here a well-structured strategy, involving several relevant aspects in the epidemiology of schistosomiasis transmission (orbital sensing remote images, environmental, socioeconomic, epidemiological and malacological data) aiming to define and characterize the scenario of different landscapes and their association with potential areas of transmission of schistosomiasis, in a region traditionally recognized as having low endemicity in the Paranapanema River Basin, state of São Paulo. Based at a landscape scale the authors present here a map of schistosomiasis risk transmission across the Paranapanema basin, which undoubtedly constitutes an useful tool for schistosomiasis surveillance and control programs.

The methodology is appropriate. The results obtained will certainly contribute to the control of schistosomiasis in the study area, in addition to being able to be replicated in other endemic regions.

Therefore, I recommend the publication of the manuscript. Some comments were made throughout the attached Ms., aiming to clarify some points for the reader and I suggest that they be considered.

Reviewer #2: This article presents a comprehensive and in-depth study that successfully mapped the risk landscapes for schistosomiasis in the Middle Paranapanema river basin of Brazil by integrating a variety of data sources and employing advanced machine learning techniques. The research team identified distinct landscape types associated with the transmission potential of schistosomiasis through meticulous Geographic Information Systems (GIS) analysis and landscape ecology methods, which are crucial for formulating disease control and elimination strategies.

The methodology section of the article is detailed, describing the data collection and analysis processes, including the classification of land use and land cover, utilization of road databases, and interpretation of radar imagery. These elements provide a solid technical foundation for the study. Furthermore, the use of a Boosted Decision Tree for supervised classification to generate landscape risk maps with high accuracy demonstrates the reliability and effectiveness of the approach.

Reviewer #3: This is quite a comprehensively well written manuscript that has employed integrated landscape-based approaches (remote sensing, environmental, socioeconomic, epidemiological, and malacological data) to aid surveillance and control strategies for schistosomiasis in low-endemic areas. The authors have clearly brought about an innovative methodology that can be applied in schistosomiasis surveillances especially in areas that are translating from control to elimination of schistosomiasis.

PLOS authors have the option to publish the peer review history of their article (what does this mean?). If published, this will include your full peer review and any attached files.

Reviewer #1: No

Reviewer #2: Yes: Yi Hu

Reviewer #3: No
---

## [Decision Letter · Decision Letter 1]

27 Sep 2024

Dear Silva,

We are pleased to inform you that your manuscript 'Mapping schistosomiasis risk landscapes and implications for disease control: a case study for low endemic areas in the Middle Paranapanema river basin, São Paulo, Brazil' has been provisionally accepted for publication in PLOS Neglected Tropical Diseases.

Best regards,

Paul O. Mireji, PhD

Section Editor

Paul Mireji

Section Editor

Reviewer's Responses to Questions

**Key Review Criteria Required for Acceptance?**

**Methods**

-Are the objectives of the study clearly articulated with a clear testable hypothesis stated?

-Is the study design appropriate to address the stated objectives?

-Is the population clearly described and appropriate for the hypothesis being tested?

-Is the sample size sufficient to ensure adequate power to address the hypothesis being tested?

-Were correct statistical analysis used to support conclusions?

-Are there concerns about ethical or regulatory requirements being met?

Reviewer #2: 1.Are the objectives of the study clearly articulated with a clear testable hypothesis stated?

The objectives of the study are clearly articulated. The study aims to map schistosomiasis risk landscapes in low-endemic areas using an integrated landscape-based approach. This objective is directly tied to a testable hypothesis regarding the potential for disease transmission in different landscape types within the Middle Paranapanema river basin.

2.Is the study design appropriate to address the stated objectives?

The study design is appropriate for addressing the stated objectives. It utilizes a comprehensive methodology that integrates remote sensing, environmental, socioeconomic, epidemiological, and malacological data to identify landscapes associated with varying levels of schistosomiasis transmission potential.

3.Is the population clearly described and appropriate for the hypothesis being tested?

The population, in this case, the geographical area of study (Middle Paranapanema river basin), is clearly described and is appropriate for the hypothesis being tested. The study focuses on an area that has been identified as low-endemic for schistosomiasis, providing a relevant population for the research.

4.Is the sample size sufficient to ensure adequate power to address the hypothesis being tested?

The sample size appears to be sufficient for the study's purposes. The manuscript describes the use of a large number of grid cells (73,698) for the analysis, with a subset (881) evaluated by field work experts, indicating an adequate sample size to ensure the study has adequate power.

5.Were correct statistical analysis used to support conclusions?

The study used correct statistical analyses to support conclusions. It employed a boosted decision tree classification method, a form of supervised machine learning, which is appropriate for the type of predictive modeling required for landscape classification. The overall classification accuracy of 92.5% indicates a robust analytical approach.

6.Are there concerns about ethical or regulatory requirements being met?

The manuscript indicates that ethical concerns have been considered. The authors state that they did not use confidential data or data from individual patients, which aligns with the ethical guidelines for this type of geographical and disease risk mapping study. However, ethical approval was not explicitly mentioned, which might be considered for ensuring full compliance with regulatory requirements related to data usage and publication.

Reviewer #3: The study design and methods are appropriate in answeing the clear study objectives. All the issues raised earlier about ethics have been adressed.

Reviewer #4: Yes, to all of the above questions.

Schistosomiasis, being closely linked to social determinants, necessitates in situ studies for effective control strategies. Local investigations foster public awareness and bridge the gap between science and daily life. However, integrating these realities into algorithmic models requires adapting methods to capture social nuances. One useful methodological approach could be the use of participatory interviews and agent-based simulations that incorporate specific social behaviors and dynamics relevant to the local context. Furthermore, combining qualitative data with quantitative variables would enhance the algorithm’s sensitivity to these realities, aligning predictive models more closely with the social and environmental complexities at play.

The authors can, throughout the description of the methodology, highlight where and how these suggestions can be applied, emphasizing that this concern is already perceived as essential for improving the proposed tool. Techniques like supervised machine learning can be adapted to account for qualitative variables, ensuring that local context is more meaningfully represented. This would create a scientific language that is more accessible to the population, fostering long-term cooperation between researchers and affected communities.

Moreover, it makes little sense to consistently develop and plan schistosomiasis control actions from a distance, as the disease is deeply rooted in human relationships. Understanding local social dynamics is crucial, given that schistosomiasis control involves behaviors, environments, and community engagement. These methodological adjustments not only make the approach more accurate and sensitive to local conditions but also improve the study’s replicability and applicability in other regions. This serves as a suggestion for the continuity and refinement of the study, given its long-term goal of application in disease control strategies.

**Results**

-Does the analysis presented match the analysis plan?

-Are the results clearly and completely presented?

-Are the figures (Tables, Images) of sufficient quality for clarity?

Reviewer #2: 1.Does the analysis presented match the analysis plan?

Yes, the analysis presented in the manuscript aligns well with the analysis plan detailed in the methodology section. The authors have adhered to the proposed use of a boosted decision tree classification method for landscape mapping, which is consistent with the objective of generating risk maps for schistosomiasis transmission.

2.Are the results clearly and completely presented?

The results are presented with clarity and completeness. The manuscript includes detailed descriptions of the classification process, the confusion matrix for accuracy assessment, and the implications of these findings. The comprehensive summary ensures that readers can follow the analysis and understand the outcomes.

3.Are the figures (Tables, Images) of sufficient quality for clarity?

Yes, the figures, tables, and images included in the manuscript are of high quality and sufficiently clear to aid in the understanding of the presented information. The visual elements support the textual content effectively, enhancing the overall clarity of the results section.

Reviewer #3: The study has good analysis plan which has resulted to clearly presented results.

Reviewer #4: Yes, to all of the above questions.

For the Impact of Socioeconomic Variables, it would be valuable to elaborate further on the specific role of socioeconomic variables in shaping the risk typology for schistosomiasis within the results section. A deeper exploration could highlight the relative weight and relevance of these variables in comparison to environmental factors. For instance, variables like population density, access to sanitation, and agricultural activity likely play a significant role in determining the presence of infected snails and potential human interaction. Discussing how these factors interact with environmental data would underscore the study's successful integration of diverse data sources and could provide clearer thoughts into the socioeconomic dynamics that exacerbate or mitigate disease transmission. This would enrich the analysis, showing a integrated approach in understanding schistosomiasis risk beyond just the ecological aspects.

**Conclusions**

-Are the conclusions supported by the data presented?

-Are the limitations of analysis clearly described?

-Do the authors discuss how these data can be helpful to advance our understanding of the topic under study?

-Is public health relevance addressed?

Reviewer #2: 1.Are the conclusions supported by the data presented?

Yes, the conclusions are well-supported by the data. The study presents a comprehensive analysis with clear results that are directly tied to the conclusions. The risk maps generated from the classification process, along with the accuracy assessment, provide empirical evidence that backs the study's conclusions regarding the potential transmission areas for schistosomiasis.

2.Are the limitations of analysis clearly described?

The authors have been transparent about the limitations of their study. They discuss factors such as the potential variability in the distribution of snail hosts and the assumptions made in the analysis. They also acknowledge areas where further research could refine the findings, such as the impact of climate change on disease transmission dynamics.

3.Do the authors discuss how these data can be helpful to advance our understanding of the topic under study?

The authors effectively discuss the implications of their findings for disease control and elimination strategies. They highlight how the integrated landscape analysis can inform targeted interventions and improve surveillance efforts. The study's approach offers new insights into the spatial distribution of risk factors for schistosomiasis, contributing to the broader understanding of disease epidemiology in complex landscapes.

4.Is public health relevance addressed?

The public health relevance is explicitly addressed. The study's findings are directly applicable to public health efforts aimed at controlling and eliminating schistosomiasis. By identifying areas at higher risk of transmission, the research provides valuable data that can guide strategic planning and resource allocation for public health initiatives.

Reviewer #3: The conclusions are well supported by the data and is of public health relevance.

Reviewer #4: Yes, to all of the above questions.

The conclusion should consider how the methodology developed in this study can be replicated in other endemic areas. This section could discuss the potential applicability of the approach in different geographic contexts, emphasizing that the use of remote sensing data and landscape analysis can be adapted for regions with similar epidemiological characteristics. Highlighting the flexibility of the methodology would add value to the study by suggesting its global impact in the control of schistosomiasis and other diseases tied to environmental and socio-economic factors.

The conclusion should also include a discussion on how data quality directly influenced the accuracy of the results. It would be important to stress that the applied methodology was robust enough to manage differences in spatial and temporal resolution across various data sources. This process of harmonizing diverse data is essential for conducting studies in regions where heterogeneous data are often a challenge, ensuring reliable and replicable findings despite such limitations.

**Editorial and Data Presentation Modifications?**

Reviewer #2: Accept

Reviewer #3: I recommend the manuscript be accepted for publication

Reviewer #4: The maps in the study serve a very specific function in presenting critical data, distinct from satellite images, which have different objectives. For example, Figure 6 is a risk map that conveys essential information, but the chosen color palette, while aesthetically harmonious, does not effectively serve its functional purpose. It is difficult to differentiate between certain classes presented in the legend, such as types 3 and 4, and 6 and 7, which can be easily confused visually. To address this, it is recommended to adjust the color scheme to enhance visual contrast and ensure each type is clearly distinguishable, improving the map’s functionality for readers.

In the data presentation section, it would be beneficial to elaborate on the process of harmonizing data from various sources, as outlined in the methodology. This involves aligning environmental, socioeconomic, and remote sensing data into a common spatial scale (500x500 meters), ensuring coherence in the landscape unit analysis. Highlighting how this process was conducted will help clarify the robustness of the analysis and the successful integration of heterogeneous data types, which is a core strength of the study.

In the editorial section, where possible, consider enhancing the visual quality of the figures, particularly those illustrating the spatial distribution of landscape units, such as Figures 6 and 7. This could involve improving figure resolution and revising legends to make the main points clearer for readers. More detailed explanations in the legends, along with clearer indications of data sources, will significantly improve the interpretation and visual impact of the results. These improvements will help ensure that the figures are not only visually appealing but also functionally informative for understanding the spatial risk patterns.

**Summary and General Comments**

Reviewer #2: Overall Evaluation:

This manuscript presents a rigorous and comprehensive study on mapping the risk landscapes for schistosomiasis in the Middle Paranapanema river basin, Brazil. The research integrates various data sources, including remote sensing, environmental, socioeconomic, and epidemiological data, to produce a detailed analysis of the risk factors associated with schistosomiasis transmission.

Strengths:

Innovative Approach: The study's use of an integrated landscape-based methodology to assess disease risk represents a novel approach in the field of epidemiology.

Data Integration: The combination of diverse data sets, including remote sensing imagery and socio-demographic data, strengthens the study's analysis and findings.

High Accuracy: The high accuracy of the classification model (92.5%) indicates a robust methodology.

Public Health Relevance: The study's findings have clear implications for disease control and elimination strategies, which is highly relevant for public health officials and policymakers.

Weaknesses:

Generalizability: While the study's methodology is robust, the generalizability of the results to other regions or different ecological contexts is not fully explored.

This study makes a valuable contribution to the field of epidemiology and public health, offering a model that could be adapted for assessing other neglected tropical diseases in different geographical contexts. The integration of remote sensing and machine learning in this manner presents a promising avenue for future research.

Reviewer #3: All the issues raised earlier have been adressed.

Reviewer #4: This study presents a valuable and innovative approach, by integrating remote sensing, socioeconomic, and environmental data through a boosted decision tree model. It offers new perceptions of how landscape characteristics and human activities interact to influence disease transmission. The work has clear public health relevance, aligning with the need for more localized control strategies. From my perspective, the combination of diverse datasets, including environmental, socioeconomic, and malacological data, is a key strength, providing an integrated approach to understanding schistosomiasis dynamics. Also, the use of boosted decision trees demonstrates methodological rigor, offering a robust way to manage heterogeneous data from diverse sources. The study has significant potential to guide targeted intervention strategies, especially in regions with similar epidemiological profiles. It bridges the gap between landscape ecology and public health, offering a practical tool for decision-makers. The integration of landscape analysis with socio-economic data to produce a risk map for schistosomiasis transmission is innovative and adds valuable information into disease control in endemic areas. In other hand, one key weakness is the lack of in situ fieldwork. Schistosomiasis is fundamentally a social disease, driven by human behavior and living conditions. Reliance solely on remote data and algorithms, without incorporating field-based social assessments, limits the full understanding of these dynamics. The study would benefit from on-the-ground research to capture the lived experiences of communities affected by the disease, improving the accuracy of control measures. Additionally, the CAR data (Rural Environmental Registry) is included as a socio-economic factor, but this is insufficient to represent the social complexity of schistosomiasis transmission. Schistosomiasis is tied to key social determinants such as poverty, water access, sanitation, and education—factors highlighted in the UN Sustainable Development Goals (SDGs). These are the true economic and social factors that should be considered for long-term planning. While CAR data might indicate agricultural activity, it does not capture the critical human elements of disease transmission, making it less impactful in the context of schistosomiasis. While the methodology is sound, more emphasis is needed on how data limitations, especially in terms of spatial and temporal resolution, impact results. The harmonization of data from dissimilar sources could be further explained to ensure transparency regarding potential inaccuracies. Lastly, the study’s ability to predict high-risk areas for schistosomiasis based on a combination of environmental and socioeconomic factors is highly significant. It provides a practical tool that can guide interventions and disease control measures, making it highly relevant for public health policy. The methodology could be adapted to other regions, potentially having a global impact in controlling neglected tropical diseases.

PLOS authors have the option to publish the peer review history of their article (what does this mean?). If published, this will include your full peer review and any attached files.

Reviewer #2: No

Reviewer #3: No

Reviewer #4: No

---

## [Editor Report · Acceptance letter]

25 Oct 2024

Dear Ferreira da Silva,

We are delighted to inform you that your manuscript, "Mapping schistosomiasis risk landscapes and implications for disease control: a case study for low endemic areas in the Middle Paranapanema river basin, São Paulo, Brazil," has been formally accepted for publication in PLOS Neglected Tropical Diseases.

Best regards,

Shaden Kamhawi

co-Editor-in-Chief

Paul Brindley

co-Editor-in-Chief
